# Nucleoli-localized KANSL2 as an epigenetic regulator of ribosome biogenesis in glioblastoma cells
Nicolás Budnik [1,13], Lucía Canedo[1,13], Agustín E. Morellato[1], Marina B. Cuenca[1,2,11], Martina Garmendia[1], Sergio Senin[1], Sebastián A. Romano[1], Zdenek Andrysik[3,4,5,6], Guillermo A. Videla-Richardson[7], Michael W. Graner[8], Ken Kobayashi[9], Yilong Zhou [10,12], Meike Wiese [10], Asifa Akhtar [10], Joaquín M. Espinosa [3,4,5] & Carolina Perez-Castro [1] ✉

KANSL2 is a subunit of the non-specific lethal (NSL) chromatin-modifying complex associated with glioblastoma (GBM) progression, but the intrinsic role of KANSL2 in GBM cells is poorly understood. By analyzing TCGA-GBM and GTEx datasets, we found that KANSL2 is upregulated in GBM and positively correlates with genes involved in ribosome biogenesis. Immunofluorescence and cell cycle analyses reveal a dynamic nuclear distribution, with KANSL2 becoming enriched in nucleoli mainly during G1/early S and G2 phases. Overexpression of KANSL2 increases 45S pre-rRNA and 28S rRNA levels, whereas its silencing reduces rRNA expression and histone H4 acetylation at lysines 5 and 8 within rDNA promoters. RNA-seq of patient-derived GBM spheroids confirms a global downregulation of ribosome biogenesis genes upon silencing of KANSL2. Together, these findings identify KANSL2 as a nuclear factor that transiently associates with nucleoli to promote rRNA transcription and ribosome biogenesis, supporting the biosynthetic and proliferative capacity of glioblastoma cells.

Glioblastoma (GBM) is the most aggressive tumor among central nervous system malignancies[1]. Patients show a high rate of relapse after treatment and a low survival rate after diagnosis (14.6 months)[1]. The lack of effective treatments reflects the aggressiveness of these tumors, largely due to their high cellular heterogeneity associated with the presence of cancer stem cells with high cell plasticity[2].

The non-specific lethal (NSL) chromatin-modifying complex regulates the expression of essential genes involved in multiple cellular processes, including chromatin structure[3], transcription, DNA repair[4], and mitochondrial gene expression[5,6]. This regulation is primarily mediated through histone H4 acetylation, which promotes chromatin remodeling and transcriptional activation[7–9]. The NSL complex is composed of the acetyltransferase enzyme MOF/KAT8 and a subset of associated subunits, including KANSL1, KANSL2, and KANSL3, among others[10]. MOF/KAT8 is also a core component of the Male-Specific Lethal (MSL) complex, best known for its role in X chromosome dosage compensation[11], however, the NSL complex functions as a distinct epigenetic regulator of transcription in mammalian cells[9].

Recently, KANSL2 was shown to sustain the functional stability of the NSL complex in non-tumor cells[8,9]. In this context, we previously described that upregulated *KANSL2* plays a critical role in the tumorigenesis of glioblastoma (GBM), enriching the GBM stem-like cell population by regulating the expression of master pluripotency factors, such as POU5F1 and NANOG. Furthermore, KANSL2-depleted GBM cells showed lower tumorigenesis, stemness, and reduced global histone 4 acetylation (H4K16)[12].

Cancer progression is known to be modulated by the interplay between ribosome biogenesis, protein production, cell proliferation, maintenance of stem cell-like properties, and the epithelial-mesenchymal transition[13]. In addition, nucleolar hypertrophy has long been considered a clinical marker of tumor malignancy[14] since the nucleolus, involved in ribosome biogenesis, plays a crucial role in tumor development and progression. Increased ribosomal (r) RNA synthesis, particularly enhanced accumulation of pre-rRNA, has been regarded as a main driver of cellular proliferation and a potential target for therapeutic intervention[13,15]. Several pluripotency factors, including POU5F1 and SOX2, were shown to bind to the rRNA promoter in mice and human embryonic stem cells[16], suggesting that regulation of ribosome biogenesis contributes to cancer stem cell characteristics[17]. Given this, it is important to mention that the roles of epigenetic factors that regulate cell stemness and plasticity during rRNA synthesis are still poorly characterized[18–20].

To further explore how KANSL2 regulates GBM properties, we analyzed *KANSL2* expression in a large set of tumor samples. Interestingly, mRNA expression was closely associated with genes involved in ribosomal biogenesis. Notably, endogenous KANSL2 localization in the nucleolus

depends on the cell cycle phase. Furthermore, KANSL2 affected histone acetylation levels at the rDNA promoter, pre-rRNA transcription, and ribosome-related proteins, revealing a KANSL2 role in rRNA biogenesis in GBM cells. Our finding that the epigenetic factor KANSL2 regulates ribosome biogenesis and stemness properties, ultimately affecting tumorigenesis, makes it a promising target for therapeutic cancer treatment.

## Results

### Upregulated *KANSL2* in GBM positively correlates with ribosomal protein expression

To better characterize the role of KANSL2 in GBM, we analyzed the expression of *KANSL2* in public RNA-seq data from TCGA-GBM and GTEX normal brain tissue[21]. The transcriptomic meta-analysis revealed that *KANSL2* expression is significantly enriched in GBM samples compared to normal brain tissue (Supplementary Fig. 1), as was previously observed in a small cohort[12]. Notably, the expression of other members of the NSL complex, including MOF/KAT8, WDR5, KANSL1, and KANSL3, was unaltered or even lower in GBM, except for MCRS1 (Supplementary Fig. 1). Analysis of the association between *KANSL2* expression and cell pluripotency in GBM, calculated as a stemness score index[22], was statistically significant (Supplementary Fig. 1) in line with our previous results. Interestingly, gene ontology analysis showed a direct association between *KANSL2* mRNA expression and the terms "ribosome" and "ribosome biogenesis" in tumor samples with high *KANSL2* expression (*$p < 0.05$) (Fig. 1a, c). Delving into the association of KANSL2 expression with the term "ribosome", gene set enrichment analysis (GSEA) showed a normalized enrichment score (NES) of 6.496 (FDR < 0.001) (Fig. 1b). These results suggest a strong association of KANSL2 with ribosomal function in GBM tumor samples.

### *KANSL2* localizes at the nucleolus

To further determine the association between KANSL2 and ribogenesis, we performed immunofluorescence (IF) assays of KANSL2 in proliferating

human glioblastoma (GBM) cell lines U87 and U251, revealing a heterogeneous nucleolus localization, in addition to nucleoplasm and cytoplasm (Fig. 2a and Supplementary Fig. 2b). Nucleoli were determined by UBF (a DNA-dependent fibrillar centers marker) and FBL (Fibrillarin, an RNA-dependent dense fibrillar component marker)[23] staining. Likewise, nucleolar localization of endogenous KANSL2 was observed in the GBM patient-derived line G08, which shows a cancer stem cell signature[12,24] (Supplementary Fig. 2a). Additionally, a similar nucleolar-staining pattern was observed in the embryonic kidney cell line HEK293T, and neural progenitor cells (NP) (Supplementary Fig. 2a), suggesting KANSL2 localizes at the nucleolus, not only in cancer cells but also in normal cells.

To determine whether KANSL2 nucleolar localization depends on active transcription of ribosomal (r) RNAs[25,26], we evaluated if pharmacological inhibition of RNA polymerase I (POL I) with actinomycin D (ACTD)[27–29] affects nucleolar localization of KANSL2 in U87 cells (Fig. 2b, c). KANSL2 and UBF nucleolar localization was determined by Pearson's correlation coefficient of pixel's intensity (referred as Correlation coefficient) (Supplementary Fig. 3) and indicated that KANSL2 nucleolar localization declines with ACTD treatment. The effectiveness of ACTD treatment was confirmed, as observed by UBF relocation in nucleolar caps[30]. Importantly, ACTD did not alter KANSL2 levels, suggesting that location of KANSL2 was altered by ACTD. To further investigate the nucleolar localization of KANSL2, the silencing of *KANSL2* in U87 by shRNA-mediated knockdown (KD) was evaluated, and showed a decrease of KANSL2 expression, including the nucleolar signal (IF) (Supplementary Fig. 4), suggesting the KANSL2 IF nucleolar labeling is specific.

### Nucleolar localization of KANSL2 changes during the cell cycle

The biogenesis of rRNA is a tightly regulated process throughout the cell cycle, ensuring proper cell growth and proliferation[31,32]. During this process, epigenetic factors influence the transcription of rRNA[33]. Although it remains uncertain whether MOF/KAT8 acts in a complex or alone, it

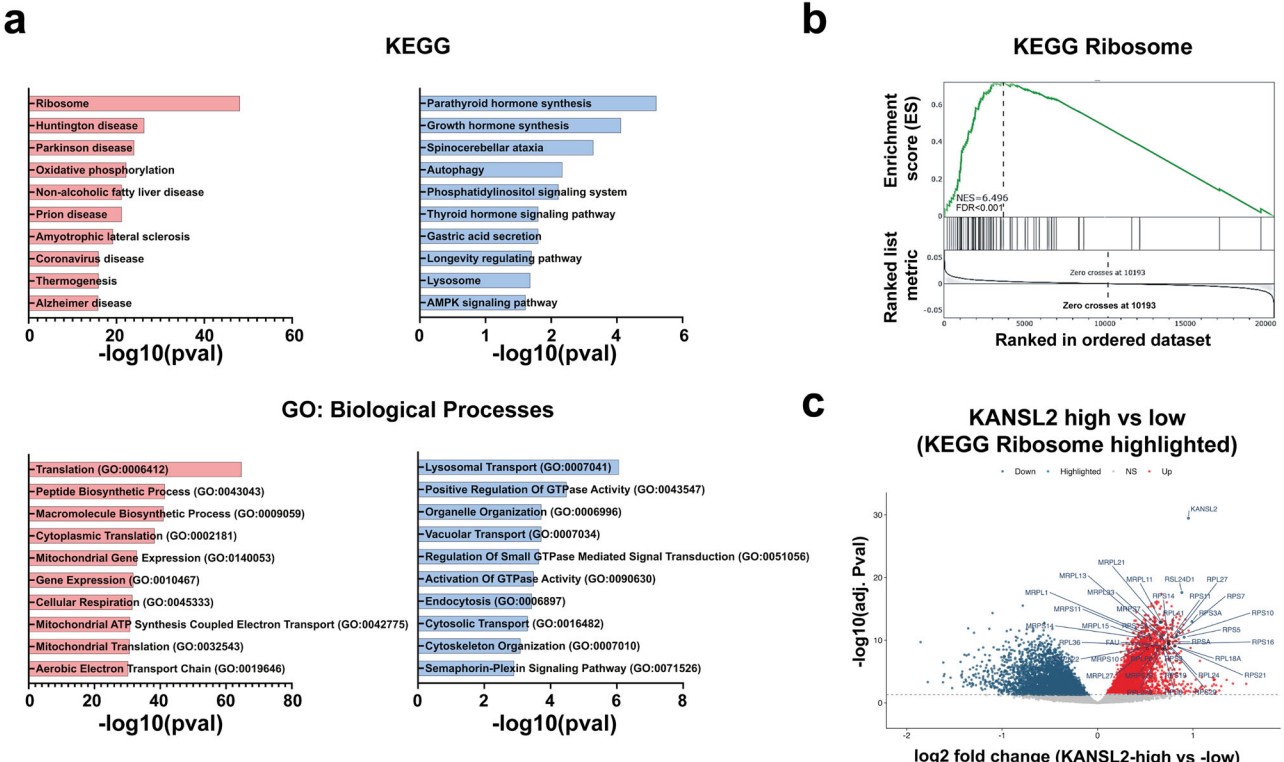

**Fig. 1 | *KANSL2* expression levels on Human GBM samples. a** Canonical pathways enrichment analysis of KEGG and GO Biological Processes of differentially expressed genes (Red: Upregulated genes; Blue: Downregulated genes) of high and low *KANSL2* expression TCGA-GBM samples. **b** Gene set enrichment analysis of high vs low *KANSL2* expression in TCGA-GBM samples. **c** Volcano plot of high vs low *KANSL2* expression in TCGA-GBM samples (KEGG Ribosome genes and KANSL2 Highlighted).

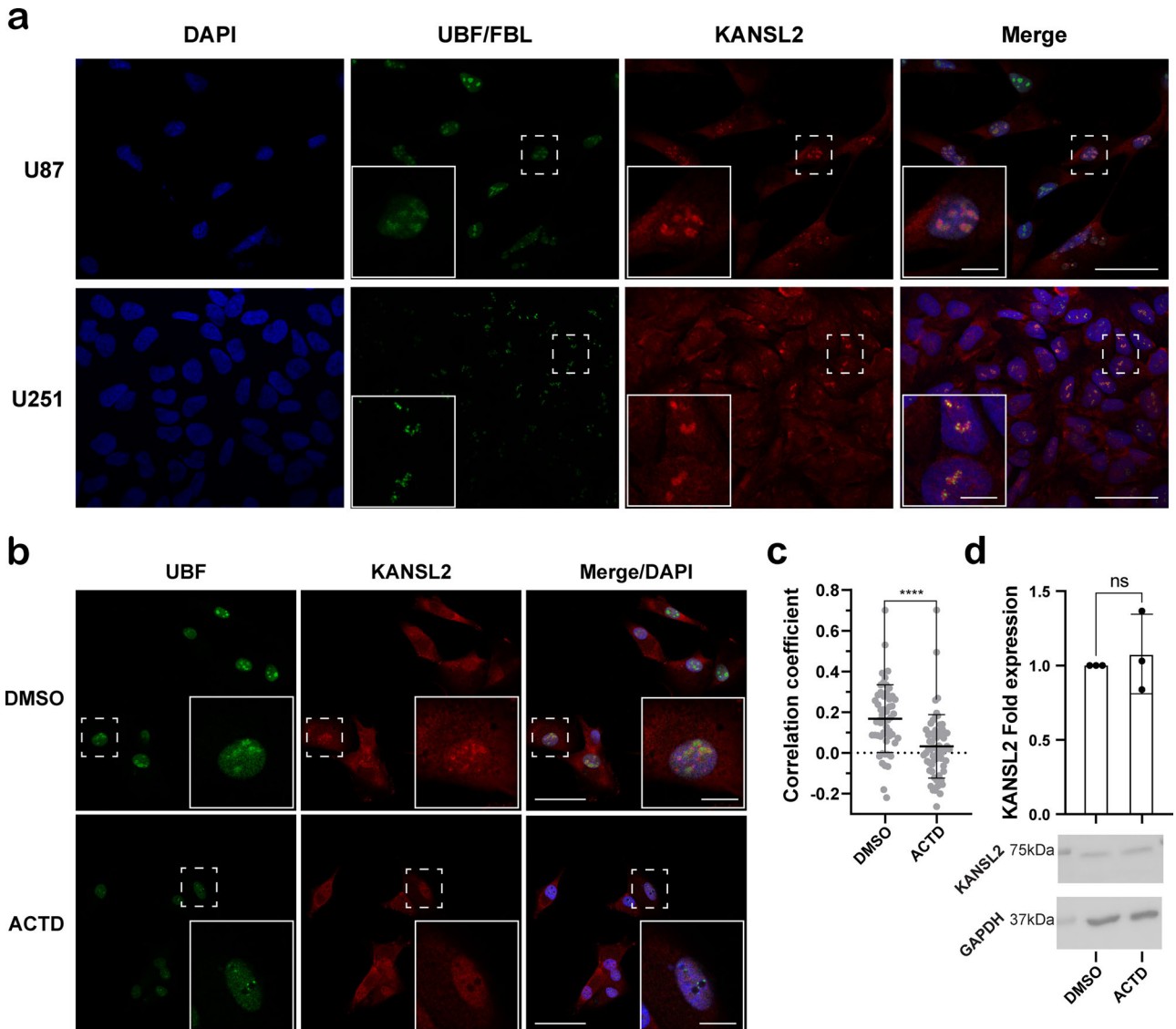

**Fig. 2 | Nucleolar localization of KANSL2. a** Immunofluorescence detection of endogenous KANSL2 and nucleolar markers UBF in U87 and FBL in U251 cells. A dashed box corresponding to the zoomed area is shown in greater detail (scale bars: 50 μm and 10 μm for insets). **b** Actinomycin D (ACTD) treatment of U87 removed KANSL2 and relocalized UBF immunofluorescence signals from nucleolus. DAPI was used to stain the nuclei. A dashed box corresponding to the zoomed area is shown in greater detail (scale bars: 50 μm and 10 μm for insets). **c** Quantitation of relative co-localization of KANSL2 in nucleolus upon ACTD treatment ($n = 54$ cells for DMSO, and $n = 64$ cells for ACTD) ($t$-test was performed; data are represented as mean ± SD; **** $p \leq 0.0001$). **d** Quantification of KANSL2 by western blot after ACTD treatment (top) and representative image of western blot (bottom) ($n = 3$; $t$-test was performed; data represented as mean ± SD).

associates with the rDNA gene in a cell cycle-dependent manner[34]. At the end of the S phase, under low-energy conditions, MOF/KAT8 inhibits rRNA expression by acetylating TIP5, a member of the epigenetic NORC complex. This acetylation modification alters TIP5's function and results in the repression of rRNA transcription[35].

Since KANSL2 is part of the NSL complex containing MOF/KAT8, we wondered whether the nucleolar localization of KANSL2 would also depend on the cell cycle. To examine this, we synchronized HeLa cells with a double thymidine pulse and quantified the nucleolar localization of KANSL2 along the cell cycle. IF revealed that nucleolar KANSL2 levels were high during the late G1/early-S phase (T0), then decreased significantly during the mid-S and late-S phase (T3 and T6) and finally recovered during the G2 phase (T9) (Fig. 3a, b). However, the levels of total KANSL2 protein determined by Western-blot remain relatively constant along the cell cycle (Fig. 3c), as similarly reported for other KANSL proteins (i.e., KANSL1 and KANSL3)[36]. Therefore, our results suggest that KANSL2 subcellular relocalization is linked to the cell cycle phase and not to downregulation.

To further characterize the dynamics of KANSL2 distribution, we expressed KANSL2-RFP fusion protein in U87 cells and evaluated its localization. Previously, this tagged protein was shown to mimic the effects of endogenous KANSL2 by regulating the expression of stemness factors in human GBM cells[12]. KANSL2-RFP accumulated at the nucleolus, although with a more diffuse distribution in the nucleoplasm compared to endogenous KANSL2 (Fig. 4a, b). Importantly, KANSL2-RFP partially co-localized with the nucleolar marker GFP-p14ARF[37], confirming the nucleolar localization of KANSL2-RFP (Fig. 4a). Additionally, cells in S phase were determined by tracking PCNA-GFP (i.e., in the mid-S phase, PCNA foci localize near the nucleolus or at the nuclear periphery)[38–40], KANSL2-RFP nucleolar levels were decreased (Fig. 4b). On the contrary, KANSL2-RFP mainly localized at nucleoli foci during the non-S phases (G1 and G2). Thus, quantification of nucleolar-localized KANSL2-RFP through non-S and S phases confirmed the dynamic localization feature of KANSL2-RFP during the cell cycle (Fig. 4c). Some nucleolar proteins involved in ribosome biogenesis such as Nucleolin, RPS6, and c-Myc shuttle between

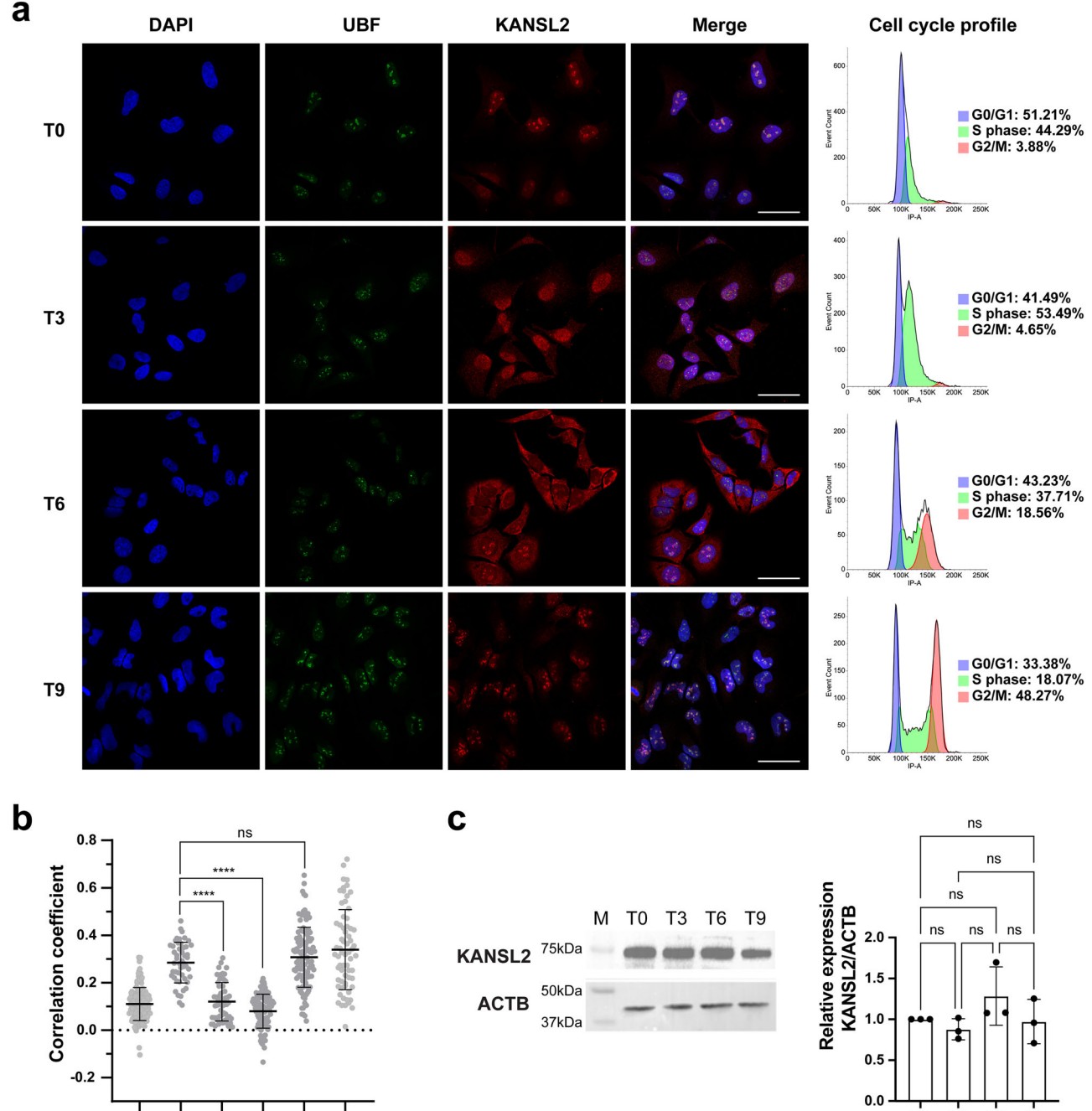

**Fig. 3 | Nucleolar localization of KANSL2 in synchronized cells.**
**a** Immunofluorescence detection of endogenous KANSL2 and nucleolar marker UBF in Hela cells. DAPI stained the nuclei. Adjustments to individual channels were performed to better visualize the merged images. (scale bar, 50 μm). Cell cycle was determined through propidium iodide staining and flow cytometry of synchronized cells. **b** Correlation coefficient (i.e., Pearson's correlation coefficient of pixels

intensity) ($n = 57$ for T0, $n = 102$ for T3, $n = 111$ for T6, $n = 132$ for T9, $n = 132$ for Negative control (NEG) and $n = 66$ for positive control (POS), respectively). For the comparative analysis, raw images were used (ANOVA followed by Dunnett´s test was performed; ****$p \leq 0.0001$; data are represented as mean ± SD). **c** Western blot analysis revealing KANSL2 protein levels in cell cycle stages (ANOVA followed by Tukey's was performed; $n = 3$; data represented as mean ± SD).

nucleolus, nucleoplasm and cytoplasm during the cell cycle[41–43], which might suggest that KANSL2 behaves similarly.

## KANSL2 modulates *MOF/KAT8* expression levels and rRNA synthesis

We previously observed that *KANSL2* knockdown (KD) inhibits *MOF/KAT8* expression in GBM cells and, consequently, could impact the activity of the NSL complex[12]. Accordingly, KANSL2-depleted cells showed global reduction of H4K16Ac, H4K8Ac and H4K5Ac (Supplementary

Fig. 5). In addition, transient knockdown of KANSL2 in U87 cells (siKANSL2) also decreased the expression of MOF and other members of the NSL complex such as KANSL1 and KANSL3 (Supplementary Fig. 6), suggesting the importance of KANSL2 for the NSL complex stability as was previously reported in mouse embryonic fibroblasts (MEFs) and podocytes[8].

The overexpression of KANSL2-RFP increased the expression of *MOF/KAT8* in U87 cells (Fig. 5a). Regarding ribogenesis, we observed an increased expression of the RNA polymerase I subunit E (*POLR1E*) (Fig. 5a),

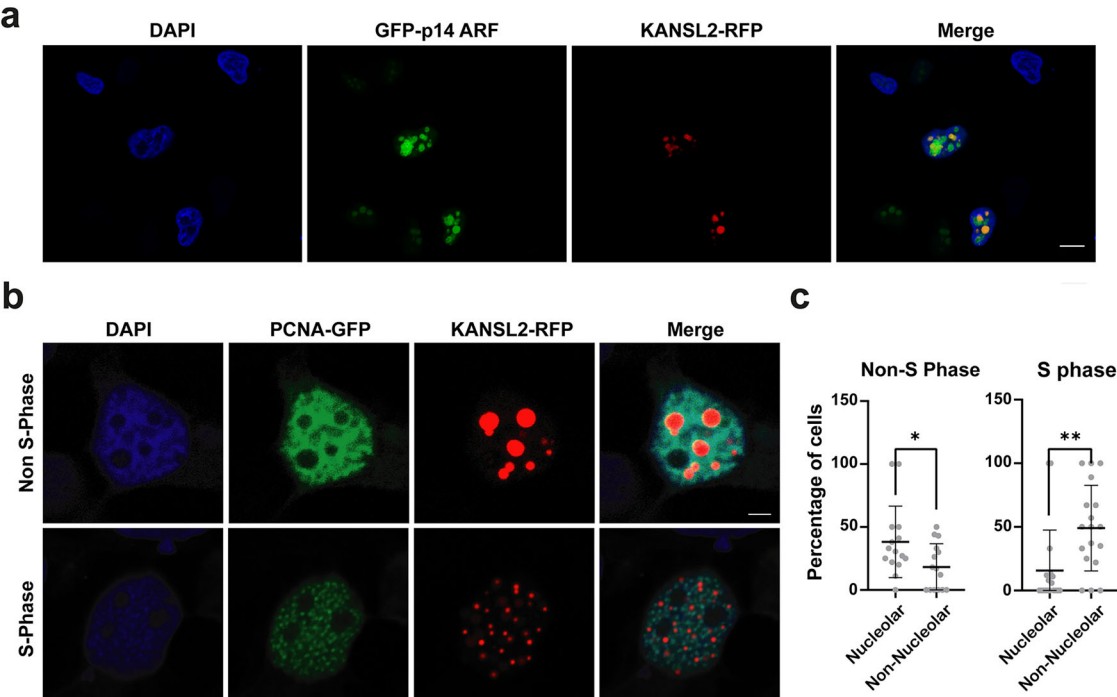

**Fig. 4 | KANSL2-RFP nucleolar localization. a** Representative images from cells overexpressing KANSL2-RFP and GFP-p14ARF. scale bar, 10 μm. **b** KANSL2-RFP and PCNA-GFP, in the S and Non-S phase. Scale bar, 5 μm. **c** Percentage quantification of nucleolar localized KANSL2-RFP. DAPI was used to stain the nuclei (*t-test* was performed; *n* = 150; * *p* ≤ 0.05, **p* ≤ 0.01; data represented as mean ± SD).

which is critical for the POL I pre-initiation complex assembly, the initiation of rDNA transcription, and consequently in cell growth[14,44]. In addition, the 45S pre-rRNA and 28S rRNA levels were also increased, except for the 18S rRNA (Fig. 5a). In turn, upon KANSL2 depletion, *POLR1E*, 45S rRNA precursor, and 28S rRNA expression levels were reduced (Fig. 5b).

Notably, when transfecting cells with a reporter plasmid driving luciferase expression from the ribosomal promoter (rDNA-luc), we observed a significant increase in luciferase activity when KANSL2-RFP was co-expressed. On the contrary, a decrease in luciferase reporter activity was measured in KANSL2 KD cells, confirming the role of KANSL2 in rRNA synthesis (Fig. 5c). In addition, U87 KANSL2 KD cells showed decreased expression of ribosome-related proteins (Supplementary Fig. 6).

To validate these findings, we silenced endogenous KANSL2 expression in GBM cells and restored its levels by expressing KANSL2-RFP (which is not the target of shKANSL2 II). Under these conditions, the KD effects were reversed, as evidenced by the increased levels of the 45S pre-rRNA and 28S rRNA (Supplementary Fig. 7), confirming the specificity of KANSL2's role in rRNA biogenesis.

To analyze whether KANSL2 could also affect ribosome function, we performed a puromycin incorporation assay into the nascent peptides, which assesses the translational capacity of cells[45]. Western-blot analysis revealed decreased puromycin incorporation in U251 KANSL2 KD cells (Fig. 5d). Similarly, puromycin labeling of KANSL2 KD cells showed decreased puromycin immunostaining in U251 (Fig. 5e), U87, and HeLa cells (Supplementary Fig. 6). These results confirmed that KANSL2 is required for correct ribosome formation and protein synthesis.

### *KANSL2* expression regulates cell proliferation and cell cycle progression in GBM cells

Based on the evidence supporting that KANSL2 positively regulates rRNA biogenesis, we hypothesized that the increased level of KANSL2 might boost cell proliferation and tumorigenesis through increased rRNA biogenesis. As expected, the overexpression of KANSL2-RFP in U87 significantly increased cell proliferation. Conversely, silencing of KANSL2 reduced their

growth, and the specificity of KANSL2 effect was confirmed by complementation assay overexpressing KANSL2-RFP in KD KANSL2 cells (Supplementary Fig. 7). Moreover, under stably depleted condition of KANSL2, propidium iodide-stained cell cycle assays confirmed a reduction of number of cells in the S phase (Supplementary Fig. 7), as evidenced by a significant accumulation of cells in G0/G1. This result aligns with previous reports, as NSL complex members have been shown to regulate cell cycle progression in HEK293T cells by affecting H4K16 acetylation on the promoters of genes critical for cell proliferation[46]. Our findings on the effects of KANSL2 on cell cycle progression in GBM cells, then, could be intriguingly linked to its role in rRNA biogenesis[47,48].

To further evaluate the mechanisms by which KANSL2 affects ribogenesis, we assessed the acetylation levels of histone 4 (H4) associated with enhanced transcription at the rDNA promoter and gene body regions (Fig. 6). By performing chromatin immunoprecipitation (ChIP) assays with antibodies directed against acetylated lysine 5 (H4K5Ac) and 8 (H4K8Ac), two previously reported acetylation marks catalyzed by the NSL complex[9], we found reduced levels of Histone 4 (H4) acetylation at the promoter region of rDNA in KANSL2-depleted cells (Fig. 6). This result suggests that KANSL2 is involved in rRNA transcription regulation through the NSL complex.

### Bulk RNA-seq data from patients-derived GBM spheroids showed KANSL2 KD affects ribosomal gene expression

To investigate the global transcriptomic profile under KANSL2 depletion, we performed bulk RNA-seq analysis of two patient-derived GBM cell lines (F18-1 and F2-4) cultured in 3D spheroids (Fig. 7a). Differential expression analysis between KANSL2 KD (shKANSL2) spheroids and the control (shNT) revealed a significant number of both increased and decreased genes (Fig. 7b, c). To reduce potential off-target effects of silencing, as well as sample-specific effects that may affect the selection of potential genes (such as low reproducibility), we selected genes that were modulated exclusively in both cell lines by two independent shKANSL2 hairpins shRNA KANSL2 I and III. Gene Ontology and KEGG signaling pathway analysis revealed a

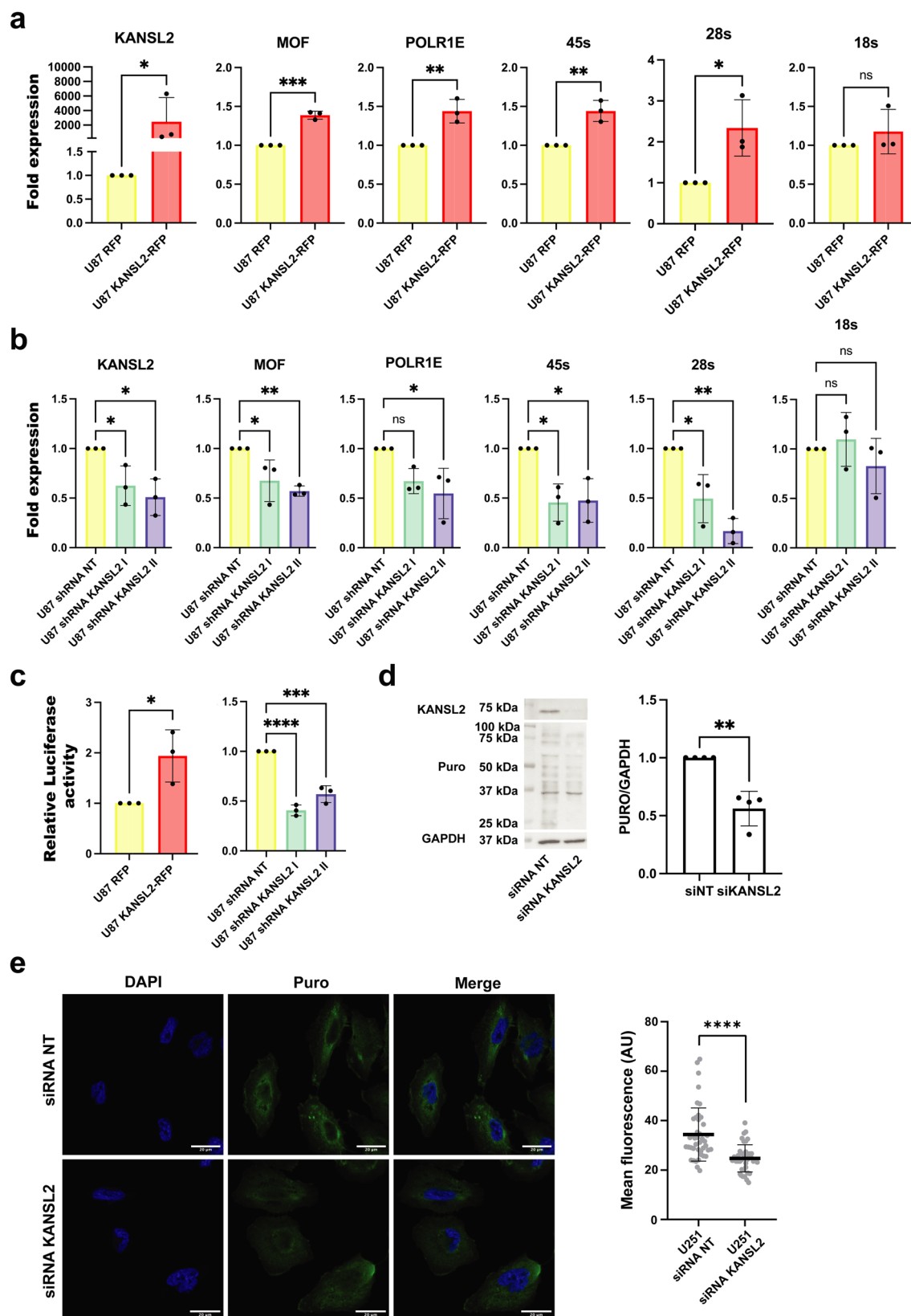

significant downregulation of genes related to ribosomal biogenesis, and mRNA processing, and ribosomal protein components (Fig. 7d, e), suggesting that KANSL2 expression is essential for ribosome biogenesis not only during rRNA synthesis but also for the transcriptional regulation of the protein components and potentially on its stoichiometric assembly.

## Discussion

In this study, we confirmed that higher *KANSL2* mRNA expression in GBM tumor samples positively correlated with pluripotency index scores, in agreement with our previous finding that KANSL2 expression is necessary for GSCs enrichment[12]. Notably, the expression of *KANSL2* was

**Fig. 5 | KANSL2 levels affect RNA biogenesis. a** qRT-PCR quantification of KANSL2, MOF, POLR1E, 45S pre-rRNA, 28S and 18S rRNAs levels in U87 KANSL2-RFP overexpressing cells. Beta-actin was used as a normalization control. (*t-test* was performed; (*n* = 3); data represented as mean ± SD). **b** qRT-PCR analysis of KANSL2, MOF, POLR1E, 45S pre-rRNA, 28S and 18S rRNAs levels in U87 GBM cells treated with two different shRNAs (shRNAs KANSL2 I and II) targeting KANSL2 mRNA. Beta-actin was used as a normalization control (ANOVA followed by Dunnett´s test was performed; (*n* = 3); data represented as mean ± SD). **c** rDNA-LUC assay in: KANSL2 overexpressing U87 cells (*t-test* was performed; (*n* = 3); *

$p \leq 0.05$; ** $p \leq 0.01$; *** $p \leq 0.001$; **** $p \leq 0.0001$; U87 KANSL2 KD cells (ANOVA followed by Dunnett's test was performed; (*n* = 3); in both cases data represented as mean ± SD). **d** Western blot analysis and quantification of pur-omyciated peptides under KANSL2 siRNA knockdown (*t-test* was performed; (*n* = 4); ** $p \leq 0.01$; data represented as mean ± SD). **e** Representative confocal images of puromycin signals (scale bar, 20 µm; left) and quantification of U251 cells siRNA NT and siRNA KANSL2 cells 48 h after transfection) treated with puromycin 10 µg/ml for 10 min and fixed (*n* = 40 for siRNA NT and *n* = 40 for siRNA KANSL2; *t-test* was performed; **** $p \leq 0.0001$; data represented as mean ± SD).

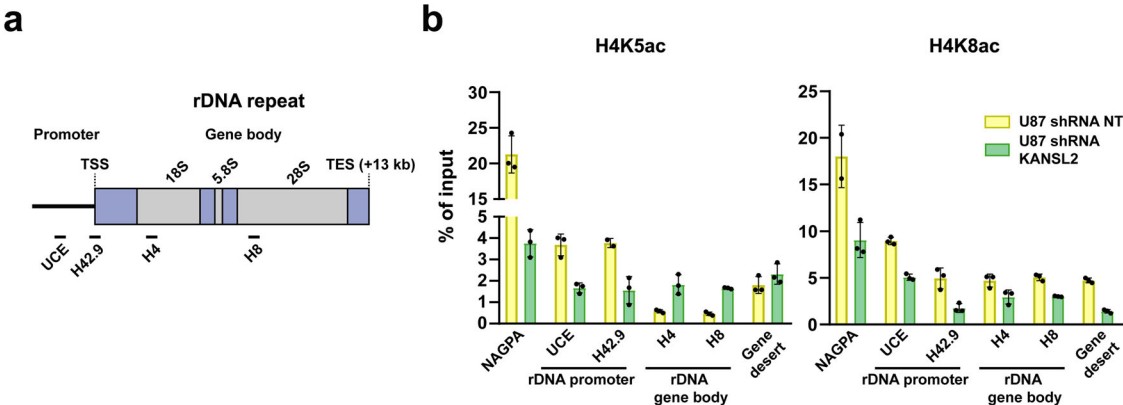

**Fig. 6 | KANSL2 downregulation reduced H4 acetylation on the rDNA promoter activity. a** Scheme of a single repeat of rDNA mapping the regions used for ChIP analysis (TSS, transcription start site). **b** qRT-PCR quantitation of H4K5ac and H4K8ac ChIP signal at selected loci after KDs of KANSL2. Values are shown as mean ± SD. (NAGPA, positive target; H42.9, and UCE, rDNA promoter-TSS; H4 and H8, rDNA gene body; Gene desert, open chromatin); (*N* = 3; technical replicates); the experiment was independently performed two times; each yielding consistent results; data from one representative independent experiment is shown.

significantly associated with the KEGG term "ribosome" in GBM samples. Subcellular localization studies revealed that KANSL2 partially localizes in the nucleolus, which may depend on basal transcriptional activity of POL I and implying a potential biological function of KANSL2 as a regulator of rRNA synthesis.

The nucleolus is an active driver of physiological cellular responses and it has been linked to processes related to tumorigenesis, stem cell maintenance, neurodegenerative disorders, and aging[26,49–51]. Thus, cancer cells typically have larger and more numerous nucleoli than somatic cells, partly due to a greater demand for ribosome biogenesis[26,52]. Moreover, Ki-67 labeling at the nucleolar organizer region indicates a hyperactive rDNA transcriptional state of altered nucleoli[52–54]. In gliomas, tumor grade correlates with morphological nucleolar alterations[26,52]. In the case of human high-grade gliomas (HGG), rRNA epi-transcriptomics (i.e., ribosomal RNA 2′O-ribose methylation) and ribosome biogenesis were distinctive regarding to IDH mutational status having different sensitivities against RNA Pol I inhibitors[55].

The identification of nucleolus-localized proteins presents a difficulty because they have intrinsically disordered domains and are located in a region of liquid-liquid phase interaction[56]. Recently, the proteome of the nucleolus revealed a large number of proteins not previously reported to occupy this location[57], evidencing that a further characterization of the nucleolus is needed. Notably, many of these nucleolus-localized proteins show spatiotemporal dynamic localization along the cell cycle or are dependent on a particular stimulus and characterized to have multiple locations within the cell. That is the case of c-Jun with nucleolar targeting that was unrecognized until recently[58]. The mechanism by which KANSL2 is recruited to the nucleolus remains unclear, although it is plausible that an interacting partner mediates its localization. Further studies are needed to elucidate this process.

Remarkably, overexpressing KANSL2-RFP resulted in partial localization in the nucleolus. This also increased steady-state rRNA biosynthesis. In contrast, silencing KANSL2 decreased rRNA biosynthesis. A luciferase activity reporter assay (using ribosomal rDNA-Luc promoter) confirmed

that KANSL2 affects transcription initiation. Moreover, reduced acetylated levels of H4 at lysine 5 and 8 (H4K5ac and H4K8ac) were observed in the rDNA promoter region in KANSL2-silenced cells. This suggests the NSL complex regulates rRNA expression epigenetically. This finding aligns with the known epigenetic status of rRNA genes during the cell cycle in synchronized HeLa cells. Acetylation of rDNA promoter-associated histones and transcription of rRNA primarily occur in late G1 and G2[59]. These are the time frames in which we observed KANSL2 localization in the nucleolus, suggesting that KANSL2 is required for, and may precede, the initiation of rRNA transcription. In summary, these studies suggest that KANSL2 mediates ribogenesis in GBM.

Ribosomal RNA (rRNA) processing is a vital cellular process and recent studies have shown that is spatially segregated within the nucleolus: early processing events leading to 18S (SSU) formation occur in the dense fibrillar component (DFC), while later stages generating 28S/5.8S (LSU) take place in the granular component (GC)[60]. Also, pre-rRNA processing and maturation regulate the shape and phases of compartmentalization observed in the nucleolus[60]. Our results indicate that KANSL2 levels also regulate the expression of enzymes and factors (mainly r proteins) required for proper ribosome processing and assembly. Particularly, KANSL2 KD caused a differential accumulation of 45S and 28S rRNA, accompanied by disregulated rRNA protein expression. Potentially, KANSL2 could specifically influence LSU-related processing or assembly factors located in the GC, which could explain why KANSL2 affected those rRNAs. Further experiments will be required to explore this.

Some questions remain open regarding the role of KANSL2 within the NSL complex in GBM. For example, the elevated mRNA levels of *MOF/KAT8* in GBM cells were not observed in GBM samples[12] (Supplementary Fig. 1). However, we confirmed that the silencing of KANSL2 in GBM cells decreased the expression of other members of the NSL complex, including MOF/KAT8, KANSL1, and KANSL3 (Supplementary Fig. 6), as was previously reported in different cell types[8]. These observations suggest that KANSL2 expression level might be rate limiting for expression of family members and may act as a critical factor in assembly of the NSL complex.

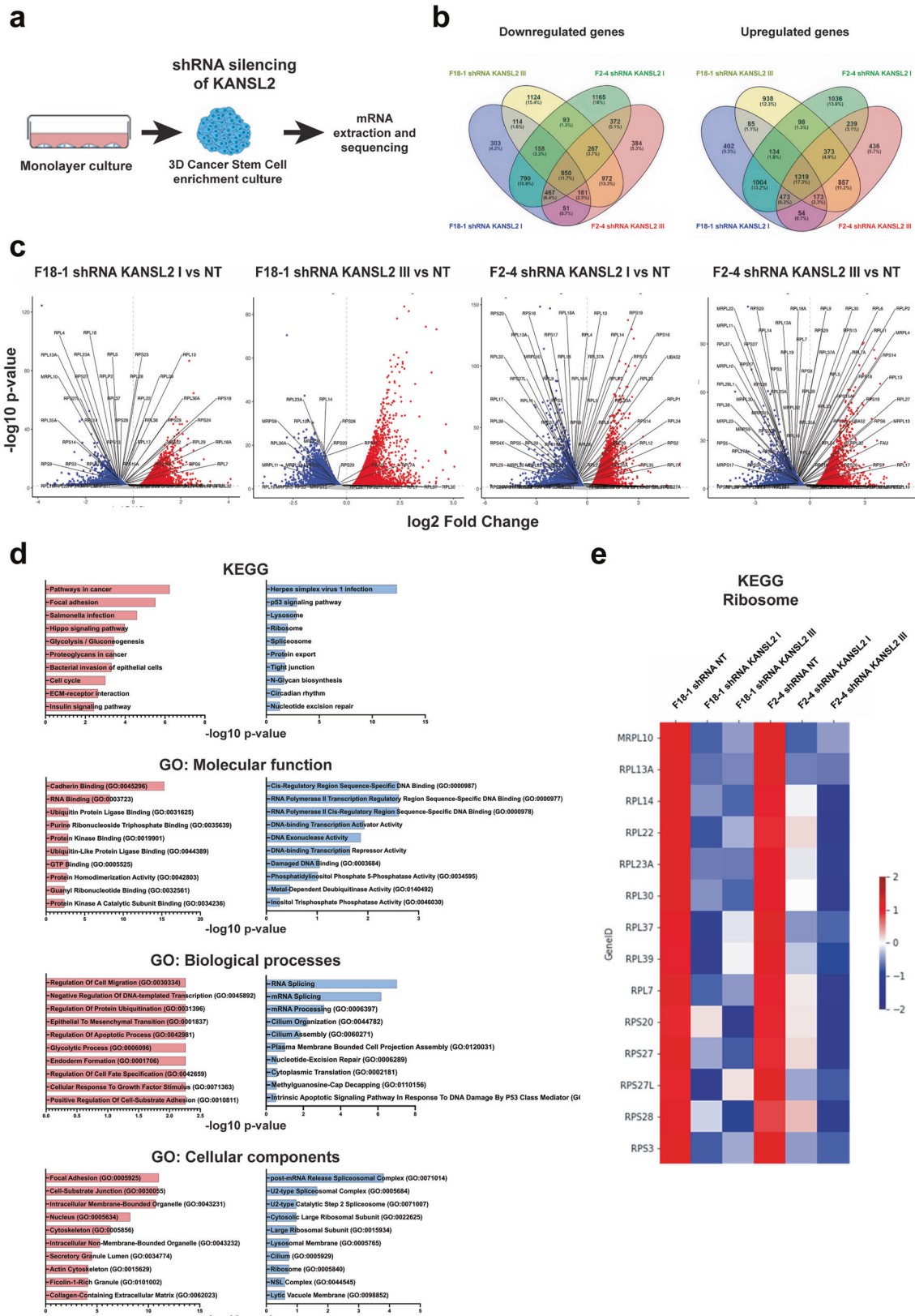

**Fig. 7 | KANSL2 KD affects ribosomal protein-encoding gene expression in GBM patients-derived spheroids. a** Schematic representation of the experimental approach to generate KANSL2-silenced spheroids with two independent shRNAs (shKANSL2 I and shKANSL2 III) from GBM grade 4 IDH-mutant astrocytoma patient-derived, F18-1 and F2-4 cells, respectively (*n* = 2). **b** Venn diagram of the number of down regulated (left) and up-regulated (right) genes. **c** Volcano plot of differentially expressed genes in the KANSL2 KD GBM derived samples. **d** Canonical pathways enrichment analysis of KEGG, GO: Molecular Functions, GO: Biological Processes and GO: Cellular Components from the differentially expressed genes (decreased in blue and increased in red) in both KANSL2 KD GBM derived samples. **e** Heat map corresponding to the expression of ribosome-related genes contained in the KEGG "ribosome" term.

When KANSL2 is in excess, MOF/KAT8 may associate with the NSL complex in place of the MSL complex to preserve essential functions in tumors[9]. It remains to be determined how MOF/KAT8 or other members of the NSL complex localize in the nucleolus and what interactions occur within the context of GBM along the cell cycle.

## Methods
Unless otherwise stated, reagents were obtained from Life Technologies or Sigma Chemical Co.

### Tissue samples
Tissue samples were collected from patients who consented to the protocol prior to resection for brain tumors surgeries in the University of Colorado Hospital/Anschutz School of Medicine Department of Neurosurgery. The tissue samples were processed and stored at the Nervous System Biorepository at the Department of Neurosurgery, University of Colorado Anschutz Medical Campus. Cell lines were grown out from these tumor tissues[61–63].

### Cell culture
The G08 cell line was derived from a human glioblastoma (GBM) grade IV biopsy, characterized previously[12,24]. NP cells were differentiated from human embryonic stem cells (WA09, University of Wisconsin; hPSCReg ID: WAe009-A)[64].

F18-1 and F2-4 cells used for the RNA-seq assay were obtained from the University of Colorado Neurosurgery Nervous System Biorepository (protocol COMIRB #13–3007). According to the WHO CNS5 (2021) categorization, F18-1 is a GBM grade 4, IDH-wildtype, MGMT promoter methylated. F2-4 is a grade 4 IDH-mutant astrocytoma (positive for IDH1 mutation by immunohistochemistry), negative for EGFR amplification, but positive for PTEN loss relative to chromosome 10 centromere sequences. F18-1 was a primary tumor and F2-4 was a recurrent resection.

Briefly, cells were cultured in serum-free medium (DMEM/F12, pH 7.2–7.4) supplemented with B27 (Thermo Fisher, prepared in buffer at pH 7.4), N2 (pH 7.4), 20 ng/ml bFGF, 20 ng/ml EGF, 2 mM L-glutamine (pH 7.4), 2 mM non-essential amino acids (pH 7.4), 50 U/ml penicillin/streptomycin, and 75 µg/ml low-endotoxin bovine serum albumin (Sigma, USA). Cells were plated onto Geltrex (A1413202)-coated plates. Media were buffered with bicarbonate and equilibrated in a humidified 5% $CO_2$ incubator at physiological pH 7.2–7.4.

U87, U251, HEK-293, and HeLa cell lines were acquired from ATCC and Merck, either directly or from colleagues, kept frozen immediately after receipt, or used in culture for less than 4 months. ATCC cell lines were authenticated by Short Tandem Repeat (STR) profiling[12] and regularly tested for mycoplasma contamination by PCR. Cells were incubated with 0.5–1 µg/ml Actinomycin D (SIGMA A9415-2MG) for 1 h in culture medium (DMEM, pH 7.4), a minimal effective dose determined previously to affect POL 1 without affecting POL 2.

### Bioinformatics analysis on patient datasets
RNA-seq expression and clinical data for glioblastoma multiforme (GBM; $n = 166$) from *The Cancer Genome Atlas* (TCGA) and normal brain tissue ($n = 1141$) from the *Genotype-Tissue Expression* (GTEx) project were obtained from the UCSC Xena browser (https://xenabrowser.net/)[21]. Harmonized datasets from the TCGA/GTEx recompute pipeline were used, provided as log2(norm_count + 1) values. All available samples were included in the analysis without further filtering.

Differential expression of *KANSL2* between GBM and normal brain tissues was assessed using a two-tailed unpaired *t*-test. To investigate the relationship between *KANSL2* and stemness features, we used the stemness index described by Malta et al.[22], specifically the mRNA expression-based index (mRNAsi), which was downloaded as precomputed values from UCSC Xena. Correlation between *KANSL2* expression and the mRNAsi was analyzed using Spearman's rank correlation coefficient.

For downstream differential gene expression (DGE), gene set enrichment analysis (GSEA), and Gene Ontology (GO) analyses within the GBM cohort, patients were stratified into "high" and "low" *KANSL2* expression groups using the mean expression value as threshold. Statistical significance was defined as $p < 0.0001$.

All statistical analyses and data visualization (boxplots and correlation scatterplots) were performed using GraphPad Prism version 10 (GraphPad Software, RRID:SCR_002798).

### RNA-seq
RNA was extracted using a QIAGEN RNeasy Kit. The integrity of the RNAs was assessed using the Agilent RNA 6000 Pico Kit and the Agilent 2100 Bioanalyzer System. The total amount of RNA sent to the company for each condition was 1 µg. The libraries were made using the NEBNext® Ultra Directional RNA Library Kit. They were sequenced with an Illumina NovaSeq 6000, 150 bp reads were paired-end by NOVEGENE, and quality control of the sample was performed using the FastQC software (RRID:SCR_014583). For the alignment with the GRCh38 reference genome (the latest available version of the Homo sapiens reference genome), the HISAT2 (RRID: SCR_015530) and SAMtools (RRID: SCR_002105) programs were used to subsequently obtain the quantification of the expression with the HTseq-count program (RRID: SCR_005514). Finally, the differential expression analysis between the silenced samples and their respective controls was performed using Bioconductor's DEseq2 software (RRID:SCR_015687).

The fastq files and pre-processed data are available in the Gene Expression Omnibus (https://www.ncbi.nlm.nih.gov/geo/) under accession number GSE291427.

### Immunofluorescence
Cells were seeded on 12-mm glass coverslips placed in 24-well plates at a density of $2 \times 10^4$ cells per well and cultured for 24 h. Cells were washed three times with PBS (pH 7.4) and fixed with 4% paraformaldehyde in PBS (pH 7.4) for 10 min at room temperature. After fixation, samples were washed three times with PBS (pH 7.4, 5 min each) and permeabilized with 0.1% Triton X-100 in PBS (pH 7.4) for 10 min. Blocking and antibody incubations were performed in PBS containing 3% BSA (pH 7.4). Primary antibodies were applied overnight at 4 °C in a humidified chamber, by placing coverslips cell-side down on 30 µl of antibody solution prepared in blocking buffer (pH 7.4). Following three PBS washes (pH 7.4), cells were incubated with the appropriate fluorophore-conjugated secondary antibodies for 30 min at room temperature. Alexa 555, 647, and 488 were used. DAPI or Hoechst 33258 was used to stain nuclei. Coverslips were mounted using Mowiol® 4-88 mounting medium (pH 8.5). All solutions were prepared in PBS adjusted to physiological pH (7.2–7.4), except for the Mowiol mounting medium.

For puromycilation assays, cells (siNT and siRNA KANSL2 cells) were incubated with puromycin (SIGMA, P4512) (10 µg/ml) for 15 min before fixing and then the coverslips were then processed for IF. Cell imaging was performed on an inverted Zeiss LSM 710 confocal microscope (Carl Zeiss Microscopy GmbH) using ZEN 2.3 SP1 Black software. 8-bit images were acquired with a 40×/1.2 NA water immersion Zeiss objective lens and QUASAR PMT spectral channels. On average, dwell time of 2.39 µs, pixel size 70–80 nm, and pinhole size 30–40 µm were used as configuration for the acquisition. The room temperature was 22 °C.

### Nucleolar localization analysis
Analysis was performed on a custom made Jupyter pipeline. Using Otsu's method for nuclei segmentation and posteriorly calculating the Pearson's correlation coefficient between the channels of interest, allowed us to quantify both channels at single cell level. The Pearson's coefficient can acquire values between -1 and 1 (in ideal negative and positive colocalization scenarios) as "Correlated coefficient" (i.e., Pearson's correlation coefficient of pixels' intensity). As reference, we performed a positive control of colocalization of the protein UBF with itself using two secondary antibodies. For

the negative control, we used UBF, localized in the nucleoli, and DAPI, which is excluded from these structures (see Extended Methods at the end of this section).

### Transfections
Cells were seeded in 6-well plates at a density of $2 \times 10^5$ cells per well and cultured for 24 h prior to transfection. Transient transfections were performed using Lipofectamine™ 3000 (Thermo Fisher Scientific) according to the manufacturer's protocol. For each well, 3.75 µl of Lipofectamine 3000 reagent was mixed with 2 µg of plasmid DNA and 4 µl of P3000 reagent in Opti-MEM reduced-serum medium (Thermo Fisher Scientific). The DNA–lipid complexes were incubated for 15 min at room temperature and then added dropwise to the cells.

The KANSL2-RFP plasmid was constructed previously[12]. GFP-PCNA is a kind gift from Dr. Vanesa Gottifredi[65]. GFP-p14ARF is a gift from Dr. Martin Monte. rDNA-LUC plasmid is a gift from Samson T. Jacob (pHrD-IRES-LUC)[66].

### Lentiviral particle generation
Knockdown cell lines were generated using Sigma Mission shRNA lentiviral plasmids targeting human KANSL2: shKANSL2 I (TRCN0000434815), shKANSL2 II (TRCN0000435099)[12], and shKANSL2 III (TRCN0000167130; sequence CGAGAGAACTTAAAGCGATTA). Mission pLKO.5-puro Non-Target shRNA plasmid (#SHC202) was used as a control. Lentiviral particles were produced in HEK293T cells seeded in 100-mm dishes at ~50–60% confluency. Cells were co-transfected with 1 µg of specific shRNA plasmid, 2 µg of pMD2.G (VSVG envelope plasmid), and 2 µg of psPAX2 (packaging plasmid, D8.9) using polyethyleneimine (PEI) at a DNA:PEI ratio of 1:6. In brief, 5 µg total DNA was diluted in 250 µl of 150 mM NaCl and mixed with 30 µg PEI diluted in 250 µl of 150 mM NaCl. The DNA–PEI complexes were vortexed, incubated for 15 min at room temperature, and added dropwise to the cells. Culture medium was replaced 4 h post-transfection. Viral supernatants were collected at 48 h and 72 h post-transfection, clarified by centrifugation ($1000 \times g$, 5 min), filtered through 0.22 µm filters, and either used immediately or stored at −80 °C.

### Viral infection and selection
Target cells were seeded in 6-well plates at a density of $1.2 \times 10^5$ cells per well. For infection, 1.5 ml of viral supernatant was added directly to each well and cells were incubated under standard culture conditions. After 6 h, the medium was replaced with fresh growth medium. A second round of infection was performed 24 h later under the same conditions. Puromycin selection was initiated 48 h after the second infection at cell line-specific concentrations (0.5–1 µg/ml) and continued every 48 h until all control cells had died.

### siRNA-mediated knockdown
Transient knockdown of KANSL2 was performed using either the Dharmacon SMARTpool: ON-TARGETplus KANSL2 siRNA (Horizon Discovery) or an individual siRNA targeting human KANSL2 (Sigma-Aldrich, SASI_Hs01_00247424). As a non-targeting control, Silencer Negative Control No. 2 (Thermo Fisher Scientific, AM4613) was used. Cells were seeded in 6-well plates at a density of $2 \times 10^5$ cells per well and transfected 24 h later with 25 pmol of siRNA and 9 µl of Lipofectamine™ RNAiMAX (Invitrogen, 13778-075), according to the manufacturer's instructions. siRNA–lipid complexes were incubated for 10–15 min at room temperature before being added dropwise to the cells. Cells were collected for analysis 48 h post-transfection.

### Cell cycle synchronization
HeLa cells were synchronized using double thymidine pulses (2 mM, T1895-1G SIGMA). First incubation with thymidine was 20 h long. Afterward, there was an 11 h release with fresh media and a second 14 h pulse with additional thymidine. This was considered the T0 of the experiment. Samples were acquired at 3 h intervals and analyzed by western

blot, immunofluorescence labeling, propidium iodide staining, and cell cytometry.

### Cell cycle phase classification with PCNA-GFP
Cells expressing the proliferating cell nuclear antigen (PCNA) fused to the green fluorescent protein (PCNA-Green) were classified by manual classification according to their PCNA nuclear distribution. Specifically, cells in the mid-S phase present foci close to the nucleoli or at the nuclear periphery; while cells in G1 and G2 have a homogeneous distribution[38,40,67–70]. Cells that did not show these characteristic PCNA features of mid-S were discarded from the analyses to avoid cells that transit between phases.

### Western blotting
Western blot (WB) analysis was performed accordingly as follows: Cells were lysed in radioimmunoprecipitation assay buffer (RIPA; 50 mM Tris-HCl pH 7.4, 150 mM NaCl, 1% Triton X-100, 0.1% SDS, 0.5% sodium deoxycholate) supplemented with a protease inhibitor cocktail (Roche), or in 2× SDS sample loading buffer (62.5 mM Tris-HCl pH 6.8, 2% SDS, 10% glycerol, 0.01% bromophenol blue, 5% β-mercaptoethanol).

Protein samples were separated by SDS-PAGE in Laemmli running buffer (25 mM Tris pH 8.3, 192 mM glycine, 0.1% SDS) and transferred to Immobilon-P PVDF membranes (Millipore) using transfer buffer (25 mM Tris pH 8.3, 192 mM glycine, 20% methanol). Membranes were blocked in TBS-T buffer (20 mM Tris-HCl pH 7.5, 150 mM NaCl, 0.1% Tween-20) containing 5% nonfat dry milk, and then incubated with primary antibodies overnight at 4 °C. After washing with TBS-T (pH 7.5), blots were incubated with goat anti-rabbit or anti-mouse HRP-conjugated secondary antibodies (Bio-Rad Laboratories, Hercules, CA, USA). Detection was performed by enhanced chemiluminescence (ECL, Supersignal, Thermo Fisher Scientific, USA).

For puromycin incorporation assays, cells (siNT and siKANSL2) were incubated with puromycin (SIGMA, P4512; 10 µg/ml) for 15 min before lysis. Antibodies used for WB are listed in Supplementary Table 1.

### Quantitative real-time PCR
Total RNA was extracted with TRIzol reagent (Invitrogen), which contains phenol/guanidine buffers adjusted to pH ~4.5 to ensure RNA integrity. cDNA was synthesized from 1 µg RNA using MMLV reverse transcriptase (Promega, USA) in the supplied reaction buffer (50 mM Tris-HCl pH 8.3, 75 mM KCl, 3 mM MgCl$_2$, 10 mM DTT). Real-time PCR was performed using the Bio-Rad CFX96 Touch™ Real-Time PCR Detection System and iTaq™ Universal SYBR® Green Supermix (Bio-Rad, USA), containing a Tris-HCl–based buffer (pH 8.4).

Beta-actin or GAPDH were used as normalization controls. Relative expression was calculated with the $2^{-\Delta\Delta CT}$ method[71]. The sequences for primers are listed in Supplementary Table 2. Averages of three independent experiments ± SD are shown.

### Chromatin immunoprecipitation (ChIP) and qPCR
Cells were fixed with 1% formaldehyde for 15 min. The fixed cells were washed twice with cold PBS (pH 7.4), scraped from the plate, and resuspended in lysis buffer (10 mM NaCl, 10 mM Tris-HCl pH 8.0, 1 mM EDTA, 0.1% SDS, and 1× protease inhibitor cocktail from Roche). Lysates were processed on a Bioruptor Plus Sonicator (Diagenode) to achieve an average DNA fragment length of 100–1000 bp. Chromatin concentrations were estimated using NanoDrop Lite (Thermo Fisher Scientific) according to the manufacturer's protocol.

DNA corresponding to 400,000 cells of the resulting lysate was diluted in 5 volumes of modified RIPA buffer (90 mM NaCl, 10 mM Tris-HCl pH 8.0, 1% Triton X-100, 0.1% sodium deoxycholate, 0.1% SDS, and 1× protease inhibitor cocktail from Roche). For immunoprecipitation reactions, 10% of the lysate was taken as input, and the remaining 90% was incubated overnight at 4 °C with the corresponding antibody.

The next day, pre-washed Protein G magnetic Dynabeads (Thermo Fisher Scientific, 10004D) were added to the reactions and incubated for 3 h

at 4 °C. The beads were then washed three times with low-salt wash buffer (150 mM NaCl, 20 mM Tris-HCl pH 8.0, 2 mM EDTA, 1% Triton X-100, 0.1% SDS) and two times with high-salt wash buffer (500 mM NaCl, 20 mM Tris-HCl pH 8.0, 2 mM EDTA, 1% Triton X-100, 0.1% SDS). Beads were resuspended in ChIP elution buffer (10 mM Tris-HCl pH 8.0, 1 mM EDTA, 1% SDS).

Samples were reverse cross-linked by adding 2 μl of proteinase K (20 mg/ml; Invitrogen, #25530049), 2 μl of RNase A (10 mg/ml; Thermo Fisher Scientific, EN0531), and 4 μl of 5 M NaCl, and incubated at 37 °C for 30 min at 800 rpm, followed by ≥2 h at 65 °C and 800 rpm on a Thermomixer (Eppendorf). DNA was then purified by ethanol/sodium acetate precipitation.

The following chromatin/antibody combinations were used: H4K5Ac and H4K8Ac. All primers used for ChIP-qPCR are listed in Supplementary Table 3.

### Proliferation assays
Cells were seeded in quadruplicate into 96 wells at a density of $1.0 \times 10^3$ cells/well. After 24 h, a quadruplicate of cells was fixed during 15 min with methanol and washed with PBS pH 7. Four days later, fixed cells were stained with crystal violet for 15 min. The absorbance at 590 nm was read and all conditions were normalized to the readings of the first day of the experiment.

### Flow cytometry
Single-cell suspension was obtained by treatment with trypsin (37 °C for 5 minutes) and washed with PBS pH 7. Then, cells were fixed with 2 ml of cold 70% ethanol, and stored on ice for 1 h. Cells were pulled-down for 2 min at 4000 rpm. The cell pellet was resuspended in 0.5 ml PBS pH 7 containing 0.25% Triton X-100 and incubated on ice for 15 min. Cells were again centrifuged for 2 min at 4000 rpm. the supernatant was discarded and, pellet resuspended in 0.5 ml PBS pH 7 containing 10 μg/ml RNase A and 20 μg/ml propidium iodide stock solution (P1304MP Life Technologies). Cells were transferred to FACS tubes and incubated at room temperature (RT) in the dark for 30 min prior to FACS analyses. Data were acquired on a FACS Canto II instrument (BD Biosciences, USA) and analyzed using Floreada.io software.

### Luciferase assays
Cells ($3 \times 10^5$ cells/well) were seeded onto 12-well plates and co-transfected with Lipofectamine 3000 (Invitrogen, USA) together with the indicated expression plasmids, their corresponding controls, and a β-galactosidase (RSV β-galactosidase) reporter. Twenty-four hours after transfection, cells were lysed in Promega Reporter Lysis Buffer (25 mM Tris-phosphate, pH 7.8, 2 mM DTT, 2 mM 1,2-diaminocyclohexane-N,N,N′,N′-tetraacetic acid). Luciferase activity was measured in a reaction buffer containing Tris-glycine (pH 7.8) and luciferin substrate. β-galactosidase activity was assayed in sodium phosphate buffer (pH 7.3) supplemented with ONPG substrate.

Total protein was quantified using a NanoDrop2000 spectrophotometer (Thermo Scientific, USA). Transcriptional activity was calculated by normalizing luciferase activity to β-galactosidase activity and expressed as mean ± SD.

### Statistics and reproducibility
Statistical analyses were performed using GraphPad Prism. The statistical test applied and the definition of *n* are specified in the corresponding figure legends. For experiments involving biologically independent samples, comparisons between two groups were performed using a two-tailed Student's *t*-test, while one-way ANOVA was used for comparisons involving more than two groups. A *p* value < 0.05 was considered statistically significant where applicable.

For RNA-seq experiment, differential expression analysis was performed using appropriate statistical methods with correction for multiple testing. Thresholds for significance and the number of biologically

independent samples are specified in the corresponding Methods and figure legends.

Biological replicates are defined as independent experiments performed on separate days using independently prepared cell cultures. The exact number of biologically independent replicates used for each experiment is indicated in the corresponding figure legends.

For experiments where statistical analysis was not applied, reproducibility was assessed by repeating experiments independently with consistent results.

### Reporting summary
Further information on research design is available in the Nature Portfolio Reporting Summary linked to this article.

**Ethical approval and informed consent.** Institutional Review Board Statement: The study was conducted in accordance with the Declaration of Helsinki and approved by the Colorado Multiple Institutional Review Board (COMIRB) of University of Colorado Anschutz Medical Campus (protocol COMIRB #13–3007) with no expiration date. All ethical regulations relevant to human research participants were followed. Informed Consent Statement: Informed consent was obtained from all subjects involved in the study. Written informed consent has been obtained from the patient(s) to publish this paper if applicable.

## Extended methods
### Nucleolar localization analysis
To characterize the nucleolar localization of KANSL2 and study its changes through the cell cycle, a Jupyter pipeline was developed for image segmentation and correlation analysis. First, a nuclei segmentation was performed in the DAPI channel based on the local Otsu's method and applying different size and shape criteria with scikit-image collection[72].

The binary nuclei mask was applied to the KANSL2 and nucleolar protein channels, and a region of interest (ROI) was cropped for each nucleus. Then, a scatter plot of pixel intensities in both channels was performed for each ROI. The Pearson's coefficient (R) in that cell was computed as follows:

$$R = \frac{\sum_i (K_i - \bar{K}) \cdot (G_i - \bar{G})}{\sqrt{\sum_i (K_i - \bar{K})^2 \cdot \sum_i (G_i - \bar{G})^2}}$$

where $KK_i$ is the intensity in the $i$ pixel in the Kansl2 channel, $GG_i$ is the intensity in the $i$ pixel in the nucleolar protein channel, and $\bar{K}\bar{K}, \bar{G}\bar{G}$ are the mean intensities.

This coefficient can acquire values between −1 and 1, and does not depend on signal levels but on the proportion and relationship between signals. In particular, -1 and 1 value represent ideal anti-correlation and correlation conditions. This coefficient is now defined as "Correlation coefficient" (i.e., Pearson's correlation coefficient of pixels' intensity). Correlation of intensities of two proteins cannot be attributed to interaction, but only to spatial coincidence. Partial localization in nucleolus due to bleed through between channels was considered null due to sequential illumination and spectral acquisition of confocal images.

Since the extreme values 1 or −1 are ideal, we designed positive and negative controls to understand the dynamic range of our analysis. We do not expect to obtain a higher value of correlation coefficient than what we would find if we were studying the colocalization of a protein with itself. Therefore, we performed a positive control by immunofluorescence analysis of UBF with two different secondary antibodies (anti mouse Alexa fluor 647 and Alexa fluor 488). As negative control, we quantified the UBF and DAPI.

## Data availability
All data required to evaluate the conclusions of this work are available in the paper and/or in the supplementary Materials (Supplementary Data 1–3). Raw data from F18-1 and F2-4 cells depleted for KANSL2 expression RNA-

seq experiments have been uploaded to GEO (Gene Expression Omnibus) under accession number GSE291427. The uncropped western blot images can be found in the Supplementary Information file.

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

## Acknowledgements

We are grateful to Dr. Maria Shvedunova (Max Planck Institute of Immunobiology and Epigenetics, Freiburg, Germany) for the training provided to N. Budnik regarding ChIP techniques. We thank the past and current members of the C. Perez-Castro lab. We thank members of the IBioBA Institute for helpful discussions and support, especially Dr. Lucas Pontel. We also thank Dr. Hernan Grecco (Departamento de Física, Facultad de Ciencias Exactas y Naturales, Instituto de Física de Buenos Aires (IFIBA), CONICET-Universidad de Buenos, Buenos Aires, Argentina) for his assistance in the colocalization analysis and calculations. We thank Alejandra Attorresi and Cora Pollak (IBioBA-MPSM-CONICET, Buenos Aires, Argentina) for their technical assistance. We are grateful to the CU Dept of Neurosurgery Nervous System Biorepository, University of Colorado, USA (https://medschool.cuanschutz.edu/neurosurgery/research-and-innovation/services/nervous-system-biorepository) for access to cell lines. This study was supported by grants from Agencia Nacional de Promoción Científica y Técnica, Argentina (ANPCyT) PICT 2016-4201, PICT 2018-02891, PICT-2021-I-A-00315, PICT 2021-0076), Consejo Nacional de Investigaciones Científicas y Técnicas (CONICET), FOCEM-Mercosur (COF 03/11), and The Pew Latin American Fellows Program for the repatriation awarded to C. Perez-Castro. This work was also supported by the Argentinian Instituto Nacional del Cáncer (INC) and Fulbright PhD Research Fellowship - awarded to L. Canedo. N. Budnik, L. Canedo and M. Garmendia were supported by CONICET- doctoral fellowship.

## Author contributions

N.B.: Material preparation, data collection and curation, writing—original draft, review and editing. L.C.: Data curation; Writing—original draft. A.E.M: Data collection and curation, Writing. M.B.C: Data curation, Writing—original draft; M.G. and S.S.: Data curation. S.A.R: Data curation, Writing. G.A.V.-R: Data curation. M. W. G: Resources, Writing—review and editing. Z.A.: Data curation; Writing—original draft. K.K: Supervision; Writing—original draft; Writing—review and editing. A.A., M.W. and Y.Z: Supervision; Writing—review and editing. J.M.E: Resources, writing—review and editing. C.P.-Castro: Conceptualization; supervision; funding acquisition; Writing—original draft; Writing—review and editing. All authors read and approved the final manuscript.

## Competing interests

The authors declare no competing interests.

## Additional information

[1]Instituto de Investigación en Biomedicina de Buenos Aires (IBioBA) - CONICET –Partner Institute of the Max Planck Society, Buenos Aires, Argentina. [2]Departamento de Física, Facultad de Ciencias Exactas y Naturales, Instituto de Física de Buenos Aires (IFIBA), CONICET-Universidad de Buenos, Ciudad Universitaria, Buenos Aires, Argentina. [3]Linda Crnic Institute for Down Syndrome, University of Colorado Anschutz Medical Campus, Aurora, CO, USA. [4]Department of Molecular, Cellular and Developmental Biology, University of Colorado Boulder, Boulder, CO, USA. [5]Department of Pharmacology, University of Colorado Anschutz Medical Campus, Aurora, CO, USA. [6]Department of Biology, Faculty of Medicine, Masaryk University, Brno, Czechia. [7]Laboratorio de Investigación Aplicada a Neurociencias (LIAN). Instituto de Neurociencias (INEU)., Fundación para la Lucha contra las Enfermedades Neurológicas de la Infancia (FLENI)-CONICET, Buenos Aires, Argentina. [8]Department of Neurosurgery, University of Colorado Anschutz Medical Campus, Aurora, CO, USA. [9]Laboratorio de Agrobiotecnología, Departamento de Fisiología, Biología Molecular y Celular (FBMC), Instituto de Biodiversidad y Biología Experimental Aplicada (IBBEA-CONICET-UBA), Facultad de Ciencias Exactas y Naturales, Universidad de Buenos Aires, Buenos Aires, Argentina. [10]Max Planck Institute of Immunobiology and Epigenetics, Freiburg, Germany. [11]Present address: European Molecular Biology Laboratory (EMBL) Barcelona, Tissue Biology and Disease Modelling, Barcelona, Spain. [12]Present address: National University of Singapore (NUS), Singapore, Singapore. [13]These authors contributed equally: Nicolás Budnik, Lucía Canedo. ✉e-mail: cperezcastro@ibioba-mpsp-conicet.gov.ar

