## [Transparent Peer Review file · Communications Biology]

Nucleoli-localized KANSL2 as an epigenetic regulator of ribosome biogenesis in glioblastoma cells.

Corresponding Author: Dr Carolina Perez-Castro

Version 0:

Reviewer comments:

Reviewer #1

(Remarks to the Author)

Comments on: Nucleoli-localized KANSL2 as an epigenetic regulator of ribosome biogenesis in glioblastoma cells.

This manuscript attempts to establish KANSL2 as a nucleolar protein and its role in ribosome biogenesis in GBM cells. The data presented in the MS is highly correlative in nature with no emphasis whatsoever on the mechanistic understanding or implication of KANSL2. Furthermore, the data needs much greater clarity and quantification., To substantiate and arrive at a hard hitting functional role and mechanism to ascertain the involvement of KANSL2. In addition to knockdown studies, analyses of specific deletion or point mutations needs to be performed.

- 1.How can 45S pre-rRNA selectively work as a potential therapeutic target?
- 2.Line#95: 'Endogenous KANSL2 shows nucleolar localization dependent on cell cycle?' what does this statement mean? the nucleolus is defined when the nucleus is in interphase stage of cell cycle, so which stage is being referred to here?
- 3.Results: It is interesting to note the positive correlation in mRNA expression of KANSL2 and ribosome/ribosome biogenesis. However, what genes categorized as ribosome or ribosome biogenesis positively correlate with KANSL2 expression? is this expression selective for GBM? i.e. is this tissue specific?
- 4.Figure#2; A, B: The images should be acquired with much greater clarity. Zeiss LSM710 is an excellent confocal microscope (assuming stable lasers), where image acquisition and clarity is excellent, provided the samples are appropriately permeabilized and stained and images are acquired ensuring high signal to noise ratio. Figure#2; A, B: (i) What is the colocalization index of KANSL2 with FBL, UBF? FBL staining in these cells is sub-optimal and does not quite reveal the classical "grape-bunch" like structure of FBL. From these images it is rather hard to conclude that there is a measurable or significant decrease in the levels of either UBF or FBL? A more rigorous quantification of the levels of FBL, UBF needs to be performed. A concentration dependent profile i.e increase in ActD concentration could be attempted on these cells to assess if relatively lower or higher concentrations of ActD alter the intensity profiles of FBL and/or UBF? Furthermore, this ActD assay is not conclusive enough to conclude the localization of KANSL2 and its role in active rRNA expression? RNA-FISH could be performed for 45SrRNA or RTPCR for 45S rRNA to carefully assess the impact of ActD on rRNA expression.
5. Line#331: Fig.3: Again KANSL2 staining from the representative images shows that this is highly diffuse. Furthermore, the extent of colocalization is also noisy and variable? for instance are the green (UBF) and red (KANSL2) compared across each of the sections across the confocal stacks? the images show that UBF and KANSL2 staining is rather different and is altered through cell cycle. Fig#3C: It is hard and extremely difficult to conclude from a single WB image, what are the levels of KANSL2 across independent biological replicates?
6. Figure#4: From the image provided, it is rather hard to discern the localization of KANSL2 with the nucleolus. KANSL2 staining needs to be accompanied by a bonafide nucleolar marker/staining. Additional assays of collecting cells at each phase of the cell cycle and performing careful IF along with quantifications and/or western blot analyses of the lysates would lend more credence to this data. What is the effect of FBL or UBF knockdown on the nucleolar localization of KANSL2 and or across cell cycle? I suggest testing the effect of Cyclin/CDK inhibitors that perturb cell cycle and then assessing the

staining/levels of KANSL2, FBL, UBF to more firmly establish the association of KANSL2 with cell cycle if any.

7. Fig#5 and 6: While these assays are interesting and potentially suggest the involvement of KANSL2 in the regulation of RNA biogenesis and cell proliferation, would be useful to attempt rescue assays using full-length constructs of KANSL2 along with specific deletion or point mutations as suggested above to address the specificity of the role of KANSL2 in RNA biogenesis, cell proliferation. These assays show strong correlation but are unfortunately not conclusive enough to arrive at their purported roles in RNA biogenesis/function. It is important to note that even minor perturbations of a multitude of nuclear or cellular proteins can impact nucleolar transcripts, since the nucleolus is a major stress sensing sub-organelle.

8. Fig,7: While it is interesting to note that there is an effect of KANSL2 Kd on genes regulating ribosome biogenesis, rescue assays would be useful to determine if overexpression of KANSL2 in an KANSL2 depleted background would rescue this effect? Is the expression dysregulation similar in cells cultured as a monolayer as opposed to a spheroid?

9. While I understand that this work focuses on GBM cells and GBM as a system to establish the role of KANSL2, the role of KANSL2 could be tested on a few other cell types to address if the effects of KANSL2 are generalizable or cell-type/tissue type specific.

Minor comments and edits:

- 1.The introduction can be written much better. There is a frequent transition between concepts. It would be more engaging to read one concept and then link this to the next concept, so that the reading is seamless and flows well.
- 2.Fig#2: Conventionally, DAPI channel is arranged to the extreme left followed by red, or green and merged.
- 3.Materials and methods; Line#133; The bioinformatics assay is rather unclear, also, would be useful to mention the statistical analyses performed?
- 4.Immunofluorescence assays: how many cells were grown and for how long? What was the pH of each of the solutions used? Image acquisition parameters: what was the X-Y frame size? voxel size?
- 5.Transfections: What was the volume of Lipofectamine 3000 used ? and for how many cells?
- 6.shRNA and siRNA conditions are incomplete, what was the concentration of the shRNA plasmid, siRNA for transfections?
- 7.pH of the buffers must be mentioned as frequently as possible for each of the methods.
- 8.Molecular weight markers are missing for most of the WB in the supplementary figures.

Reviewer #2

(Remarks to the Author)

The authors have tried to investigate the role of KANSL2, a subunit of the non-specific lethal (NSL) chromatin-modifying complex, in glioblastoma (GBM). The authors demonstrate that KANSL2 is overexpressed in GBM (compared to normal brain tissue) and is positively correlated with stemness and ribosomal protein gene expression. The protein localizes dynamically to the nucleolus, and its overexpression enhances, while its knockdown suppresses, rRNA transcription and cell proliferation. However, a major message from this paper regarding the regulation of KANSL2 and its role in stemness has been published by the same group previously, which they have acknowledged generously as ref. 19 in the manuscript. Apart from this, the only aspect of novelty is the “ribosomal” involvement where again it is not clear, whether the focus of authors is “ribosomal RNA (rRNA) or ribosomal proteins (ribosomal protein mRNA)” and this needs further experimentation to sound credible.

While the manuscript provides intriguing observations, several major and minor issues require clarification or additional experimentation to support the conclusions.

Major Comments

1. Redundancy of Initial Observations:

- a) The findings in Figure 1a and 1b replicate previously published data by the same group (ref. 19), particularly the association of KANSL2 with stemness. Presenting this as a main figure seems unjustified unless new insights are added.
- b) Moreover, the method of patient stratification based on KANSL2 expression for differential gene expression (DGE), GSEA, and GO analyses lacks clarity. The thresholding approach for “high” vs. “low” KANSL2 expressers must be described in detail. Was a statistical method (e.g., median split, tertiles, or clustering) employed, and is it biologically meaningful or relevant? Also, Fig. 1a does not indicate a bimodal expression distribution, further complicating justification for this classification.

2. Ambiguity in Localization Claims:

- a) The conclusion that actinomycin D (ACTD) treatment leads to a loss of nucleolar localization is premature unless total KANSL2 expression (both RNA and protein) is shown to remain unaffected by treatment. A decrease in signal could simply reflect downregulation, not redistribution.
- b) Similarly, the interpretation of shRNA-mediated knockdown as a loss of nucleolar localization is flawed. A global decrease in KANSL2 across all compartments is expected with knockdown, and cannot be interpreted as redistribution. Figure S3b clearly suggests a uniform reduction rather than specific relocalization.

3. Cell Cycle–Dependent Localization:

The proposed relationship between KANSL2 nucleolar localization and the cell cycle remains vague. The authors should:

- a. Quantify KANSL2 levels in nuclear vs. cytoplasmic/nucleolar fractions across different cell cycle phases.

- b. Revisit the assertion in line 358 linking cell cycle phase (particularly S-phase) with rRNA biogenesis via KANSL2.
- c. Clarify whether KANSL2 localization changes precede or follow changes in rRNA expression.
- d. Notably, the lack of change in total KANSL2 levels during the cell cycle makes this hypothesis speculative without subcellular fractionation data.

4. Selective rRNA/Ribosomal Upregulation:

The upregulation of 45S pre-rRNA upon KANSL2 overexpression without a corresponding increase in mature rRNA species (e.g., 18S, 5.8S) is puzzling. The authors should investigate and explain:

- a. Why only a subset of rRNA or ribosomal proteins (e.g., only RPS18 in Fig. S5) are affected.
- b. Whether this implies incomplete processing of rRNA or post-transcriptional bottlenecks.
- c. Additional evidence (e.g., northern blot or qPCR of intermediate species) could help resolve this.

5. Potential Contradiction in Cell Cycle Data:

The data presented in Figures 3, 4c, and 6c raise interpretational challenges:

- a. KANSL2 is reported to be low in the S-phase nucleolus.
- b. Yet, KANSL2 knockdown reduces the S-phase population.
- c. This appears counterintuitive—does KANSL2 promote or suppress S-phase entry?
- d. The authors should clarify this relationship, potentially exploring whether nucleolar KANSL2 levels act as a checkpoint or modulator for S-phase progression.

Minor Comments

1. Western Blot Quantification: All western blot results (particularly Fig. 5d) must be quantitatively analyzed with densitometry, normalized to loading controls (e.g., GAPDH). As it stands, the observed changes are marginal and may not be statistically significant.
2. Redundant Literature Findings: Results presented in Figures 6a and 6b have been reported in previous literature, as acknowledged by the authors. These should be moved to the supplementary section to preserve the novelty and focus of the main text.
3. Inconsistency in Knockdown Effects: The phenotypic outcomes of KANSL2 knockdown using shRNA I and II (Fig. 6c) show considerable variability. The discrepancy is large enough that statistical comparison between the two might reveal significant differences. The authors should address:
 - a. The knockdown efficiency of both constructs.
 - b. Whether off-target effects or differential knockdown kinetics could explain this variation.
 - c. Replicating the experiments with rescue constructs could help validate specificity.

Version 1:

Reviewer comments:

Reviewer #1

(Remarks to the Author)

While the manuscript has indeed improved after revision, my previous concerns regarding the clarity of the images of KANSL2 localization still remains as the localization is still rather heterogeneous and diffuse. In silico analyses and expression studies suggest the involvement of KANSL2 in regulating gene expression of ribosomal transcripts, the evidence is rather circumstantial in nature and does not quite address the specificity of KANSL2 localization in the nucleolus.

1. Figure 2: Please provide zoomed in images of Fig. 2A or 2B, such that the localization of KANSL2 can be clearly discerned. Since the KANSL2 panel in Fig.2A is low magnification, the staining of KANSL2 (red) is unclear. I suspect this to be due to issues with permeabilization. Cells can be pre-permeabilized with CSK buffer to partially digest the cytoplasm, which would allow better permeability of both UBF and KANSL2 antibodies, as their access especially into the nucleus and nucleolus is particularly difficult.
2. Fig.2B shows KANSL2 localization potentially in the nucleolus of only a single cell, which I find particularly worrisome since if the staining was ubiquitous and consistent, this should have shown up across the nucleolus of all cells. I suspect that the relatively low correlation co-efficient of <0.2 as quantified in Fig;2c largely stems from the rather reduced localization of KANSL2 across cells in the nucleolus, possibly due to accessibility issues. Is this correct?
3. Fig.2D: Furthermore, while the ActD treatment is appropriate, the already reduced/subdued localization of KANSL2, does not really leave much for the ActD to act upon? previous studies employing ActD treatment followed by nucleolin staining, shows a major disruption and dispersal of the nucleolus across the nucleoplasm.
4. Would be extremely useful to show zoomed inserts of the nucleus before and after KANSL2.
5. Also, the manuscript would greatly benefit by some expert editing as I notice number of typographical and grammatical errors.

Version 2:

Reviewer comments:

Reviewer #1

(Remarks to the Author)

Yes, I am satisfied with the revisions and recommend the manuscript for publication.

Rebuttal letter

We would like to thank both reviewers for their thorough review of our article. We have done our best to address most of the comments and hope you find our results convincing.

In the Abstract section, we have replaced it with:

“KANSL2 is a subunit of the non-specific lethal (NSL) chromatin-modifying complex associated with glioblastoma (GBM) progression, but the intrinsic role of KANSL2 in GBM cells is poorly understood. By analyzing TCGA-GBM and GTEx datasets, we found that KANSL2 is upregulated in GBM and positively correlates with genes involved in ribosome biogenesis. Immunofluorescence and cell cycle analyses revealed a dynamic nuclear distribution, with KANSL2 becoming enriched in nucleoli mainly during G1/early S and G2 phases. Overexpression of KANSL2 increased 45S pre-rRNA and 28S rRNA levels, whereas its silencing reduced rRNA expression and histone H4 acetylation at lysines 5 and 8 within rDNA promoters. RNA-seq of patient-derived GBM spheroids confirmed a global downregulation of ribosome biogenesis genes upon silencing of KANSL2. Together, these findings identify KANSL2 as a nuclear factor that transiently associates with nucleoli to promote rRNA transcription and ribosome biogenesis, supporting the biosynthetic and proliferative capacity of glioblastoma cells.”

Reviewers' comments:

Reviewer #1 (Remarks to the Author):

Comments on: Nucleoli-localized KANSL2 as an epigenetic regulator of ribosome biogenesis in glioblastoma cells.

This manuscript attempts to establish KANSL2 as a nucleolar protein and its role in ribosome biogenesis in GBM cells. The data presented in the MS is highly correlative in nature with no emphasis whatsoever on the mechanistic understanding or implication of KANSL2. Furthermore, the data needs much greater clarity and quantification. To substantiate and arrive at a hard hitting functional role and mechanism to ascertain the involvement of KANSL2. In addition to knockdown studies, analyses of specific deletion or point mutations needs to be performed.

Reply: We thank the reviewer for this constructive comment. We believe that our findings are unlikely to be misinterpreted when the results are evaluated as a whole. The strength and consistency of the presented evidence support the conclusion that KANSL2 regulates ribosome biogenesis in GBM.

The evidence we collected demonstrates, for the first time, that KANSL2 is a key regulator of ribogenesis and cell proliferation in GBM, extending its previously established roles in stemness and tumorigenesis. These findings clarify how KANSL2 contributes to GBM tumorigenicity and why it is increased in patient samples, suggesting new potential therapeutic strategies.

We consider that inquiring the mechanistic insight of KANSL2 (e.g. how KANSL2 enters into nucleolus, etc.), which is completely unknown, is far beyond the scope of our manuscript. However, because KANSL2 is a recruiter and assembler of NSL complex, we demonstrated that KANSL2 is involved in the epigenetic modification of rDNA promoter region, which is,

mechanistically relevant for ribogenesis. However, the analysis of specific deletion or point mutation of KANSL2 is far beyond the scope of our manuscript and can be pursued in future studies. Nevertheless, we agree that the complementation assays are necessary to strengthen our conclusions. Please find the new supplemental Figure S7.

Supplemental Figure S7 a) RT-qPCR analysis of 45S pre-rRNA, 28S, and 18S rRNA levels in U87 cells expressing the indicated constructs: non-targeting control (NT) + RFP, NT + KANSL2-RFP, shKANSL2-II + RFP, and shKANSL2-II + KANSL2-RFP. Re-expression of KANSL2-RFP restored rRNA synthesis to levels comparable to control cells in KANSL2 knockdown cells. Data are shown as mean \pm SD (n = 3). Two-way ANOVA followed by Fisher's least significant difference (LSD) post hoc test (uncorrected); *p \leq 0.05, ***p \leq 0.001. **b)** Crystal violet proliferation assay in U87 cells with stable KANSL2 knockdown or overexpression.

Data are shown as mean \pm SD (n = 3). One-way ANOVA followed by Dunnett's test; ***p \leq 0.001. **c)** Crystal violet proliferation assay in U87 and U251 cells expressing the indicated constructs: NT + RFP, NT + KANSL2-RFP, shKANSL2-II + RFP, and shKANSL2-II + KANSL2-RFP. Re-expression of full-length KANSL2-RFP rescued the proliferation defect caused by KANSL2 depletion. Data are shown as mean \pm SD (n = 3). Two-way ANOVA followed by Fisher's LSD post hoc test (uncorrected); **p \leq 0.01, ***p \leq 0.001, ****p \leq 0.0001. **d)** Cell-cycle analysis of U87 cells after KANSL2 depletion, determined by propidium iodide staining and flow cytometry. The percentage of cells in G0/G1, S, and G2/M phases is shown. Data are shown as mean \pm SD (n = 3). One-way ANOVA followed by Dunnett's test; ****p \leq 0.0001.

1. How can 45S pre-rRNA selectively work as a potential therapeutic target?

Reply: We appreciate the comment. The original reference 5 (Nguyen, L. X. T et al., 2015) describes pre-rRNA as a driver of proliferation and the therapeutic target, but does not specify the 45S. We have removed "45S" accordingly (Page 4, line 91).

2. Line#95: 'Endogenous KANSL2 shows nucleolar localization dependent on cell cycle?' what does this statement mean? the nucleolus is defined when the nucleus is in interphase stage of cell cycle, so which stage is being referred to here?

Reply: We thank the reviewer for this comment. The previous phrase #95 meant that KANSL2 nucleolar localization changes during the cell cycle phase, specifically during interphase. As shown in Figure 3 (please see below in reply 5), KANSL2 localizes in the nucleolus during the late G1/S and G2 (T9) phases, whereas in the mid-S phase and late-S phase (T3 and T6), its localization is more limited.

We replaced this statement with "Endogenous KANSL2 localization in the nucleolus depends on the cell cycle phase" (Page 4, Line 99).

These experiments are reinforced by those obtained through KANSL2-RFP transfection (Fig 4), showing KANSL2 primarily localization in the nucleoplasm during the S phase of the cycle.

3. Results: It is interesting to note the positive correlation in mRNA expression of KANSL2 and ribosome/ribosome biogenesis. However, what genes categorized as ribosome or ribosome biogenesis positively correlate with KANSL2 expression? is this expression selective for GBM? i.e. is this tissue specific?

Reply: We appreciate the reviewer for this constructive comment. We have included those ribogenesis-related genes that positively correlate with KANSL2 expression (Figure 1c, supplemental information table TCGA GBM) and also is shown in Fig. 7e and 7c.

C**Ribosomal subset and KANSL2 highlighted in TCGA-GBM**
Figure 1. *KANSL2* expression levels on Human GBM samples. c) Volcano plot of high vs low *KANSL2* expression in TCGA-GBM samples (KEGG Ribosome and *KANSL2* Highlighted) (LFC = \log_2 fold change).

Figure 7. KANSL2 KD affects ribosomal protein encoding gene expression in GBM patients-derived spheroids. a) Schematic representation of the experimental design to generate KANSL2-silenced spheroids with two independent shRNAs (shKANSL2 I and shKANSL2 III) from GBM and grade 4 IDH-mutant astrocytoma patient-derived, F18-1 and F2-4 cells, respectively. **b)** Venn diagram of the number

of down regulated (left) and up regulated (right) genes. **c)** Volcano plot of differentially expressed genes in the KANSL2 KD GBM derived samples (LFC = log₂ fold change). **d)** Canonical pathways enrichment analysis of KEGG, GO: biological processes, GO: Molecular functions and GO: Cellular components from the differentially genes (decreased in blue and increased in red) expressed in both KANSL2 KD GBM derived samples. **e)** Heat map corresponding to the expression of ribosome-related genes contained in “ribosome” term.

Since we based our analysis on public transcriptomic data from GBM, the list is tissue-specific. KANSL2 is highly expressed in GBM, among other tumors, and our previous observations indicated that KANSL2 has a tumorigenic role in this cancer type. KANSL2 shows nucleolar localization in other cancer cells as well as in neural progenitors. Based on this, we speculate that KANSL2 regulates ribogenesis in other cell types, but further experiments are needed to confirm this.

4. Figure#2; A, B: The images should be acquired with much greater clarity. Zeiss LSM710 is an excellent confocal microscope (assuming stable lasers), where image acquisition and clarity is excellent, provided the samples are appropriately permeabilized and stained and images are acquired ensuring high signal to noise ratio. Figure#2; A, B: (i) What is the colocalization index of KANSL2 with FBL, UBF? FBL staining in these cells is sub-optimal and does not quite reveal the classical "grape-bunch" like structure of FBL. From these images it is rather hard to conclude that there is a measurable or significant decrease in the levels of either UBF or FBL? A more rigorous quantification of the levels of FBL, UBF needs to be performed. A concentration dependent profile i.e increase in ActD concentration could be attempted on these cells to assess if relatively lower or higher concentrations of ActD alter the intensity profiles of FBL and/or UBF? Furthermore, this ActD assay is not conclusive enough to conclude the localization of KANSL2 and its role in active rRNA expression? RNA-FISH could be performed for 45SrRNA or RTPCR for 45S rRNA to carefully assess the impact of ActD on rRNA expression.

Reply: We thank the reviewer for this comment. However, we disagree that our images were taken inappropriately for publication. We ensured minimal artifacts during the procedure and our standardized protocol incorporates optimal controls in it. Also, we have experience in this type of experiment. Therefore, with no doubt, our images have a lower chance of artifacts during acquisition and are of high quality and clarity. All settings and details of the microscopy images, Fig. 2A and B, and Fig. 3, will be provided in the original file format with the metadata information from the microscopy to the reviewers, so that any manipulation or technical issue can be detected.

The previous Figure 2 shows endogenous KANSL2 detection in the nucleolus. UBF and FBL are considered as bona fide controls, and there is no reason to quantify them, as suggested by this reviewer. However, we have now replaced the 2a showing a better image and add a new Supp. Figure S2b. We have also included a Supp Figure S3 with the explanation of Correlation efficient (i.e. Pearson correlation coefficient of Pixels intensity) for U87 cells labelled with KANSL2 and UBF and U251 cells labelled with KANSL2 and FBL. Also we included in the supplemental data, a table (see below) with the values of the correlation coefficient for U87 and U251 (from Fig. 2a) and from the cells under ACTD treatment (from Fig. 2b).

Figure 2. Nucleolar localization of KANSL2. a) Immunofluorescence detection of endogenous KANSL2 and nucleolar markers UBF in U87 and FBL in U251 cells. Scale bar, 50 μ m. **b)** Actinomycin D (ACTD) treatment of U87 removed KANSL2 and relocalized UBF immunofluorescence signals from nucleolus. DAPI was used to stain the nuclei. Scale bar, 10 μ m. **c)** Quantitation of relative co-localization of KANSL2 in nucleolus upon ACTD treatment (control:54 cells, ACTD: 64 cells). t-test **** $p \leq 0.0001$. **d)** Representative image of western blot (left) and quantification of KANSL2 by western blot after ACTD treatment. **d)** Quantification of KANSL2 by western blot after ACTD treatment t-test (n=3).

b) Immunofluorescence detection of endogenous KANSL2 and FBL in U251 GBM cells. DAPI was used to stain the nuclei Z-Stack of the orthogonal image. Scale bar, 10 μ m.

Supplemental Figure S3: a) Flowchart describing the quantification of KANSL2 nucleolar localization, starting from raw multi-channel immunofluorescence images (KANSL2, UBF, DAPI and bright field), applying DAPI-based binary mask, isolating regions of interest (ROIs) and obtaining Pearson correlation coefficient values. b) Correlation coefficient (i.e Pearson's correlation coefficient of pixels intensity) (total number of cells for U87 UBF-K2 =108; U251 FBL-K2 =86).

Moreover, we disagree about the reviewer comments suggesting a suboptimal staining of FBL. Our images appear very similar to those reports describing FBL staining in nucleolus (see references. <https://doi.org/10.15252/embr.202256230>, doi: 10.1038/srep16495.) Please take into consideration that these cells are GBM-derived cell lines and, therefore, the shape of FBL staining may differ slightly. Nevertheless, we have now included a new picture of FBL (Fig. 2b) and KANSL2-stained Z-stack images (Supplemental Figure S2) of U251 cells for a better visualization of nucleolus staining (see above).

We did not quantify the levels of FBL and UBF in these cells, as suggested by this reviewer, since variation of FBL or UBF was not reported in the previous publications.

Treatment with ActD with a low dose is widely accepted as a means to inhibit RNA Pol1 (Burger K et al, 2010; Potapova TA, et al. 2023;12:RP88799; Shav-Tal, Y. et al. 2005). We tested between 0.5 and 1 $\mu\text{g/ml}$ of ActD, and we observed a similar effect (data not shown). Under these conditions, UBF localizes to nucleolar caps as we describe in our manuscript, conclusively indicating that it exerts an inhibitory effect on ribogenesis. Under ActD treatment, KANSL2 localized outside the nucleolus (still in the nucleoplasm). Importantly, the level of KANSL2 remains stable under ActD treatment, confirming that its expression is not altered but causes its subcellular relocation (at the max doses tested: 1 $\mu\text{g/ml}$), Figure 2c (see also a figure with western blots below).

- Burger K, Mühl B, Harasim T, et al. Chemotherapeutic drugs inhibit ribosome biogenesis at various levels. *J Biol Chem.* 2010;285(16):12416-12425.

- Potapova TA, Unruh JR, Conkright-Fincham J, et al. Distinct states of nucleolar stress induced by anticancer drugs. *Elife*. 2023;12:RP88799.

- Shav-Tal Y, Blechman J, Darzacq X, Montagna C, Dye BT, Patton JG, Singer RH, Zipori D. Dynamic sorting of nuclear components into distinct nucleolar caps during transcriptional inhibition. *Mol. Biol. Cell* 16, 2395–2413, 10.1091/mbc.E04-11-0992 (2005).

Western blot and quantification of KANSL2 by densitometry after ACTD treatment.

The quantification of the Pearson correlation coefficient (now named “Correlated coefficient”, page 7, line 192) between KANSL2 and UBF is shown in Figure 2b (as a control for ACTD treatment). As mentioned above, we also included the correlation coefficient for KANSL2 and FBL in U251 cell. This quantification method was already detailed in the extended method section. However, we further provided more details and examples of how we quantified the pixels of the images (Supp Fig. S3).

Supplemental data information. Calculated values of correlation coefficient for each figure:

Fig2c							
	n	mean	SD	SEM	Statistics		
DMSO	54	0,1683	0,1661	0,01959	Test	Unpaired t-test	
ACTD	63	0,03219	0,1555	0,02261	p-value	DMSO vs ACTD	<0.0001 (****)
Fig3b							
	n	mean	SD	SEM	Statistics		
NEG	132	0,1101	0,006054	0,006054	Test	ANOVA (Dunnett's multiple comparisson)	
T0	57	0,2848	0,08531	0,01219	p-value	T0 vs T3	<0.0001 (****)
T3	102	0,119	0,0808	0,0107		T0 vs T6	<0.0001 (****)
T6	111	0,07964	0,07154	0,007084		T0 vs T9	0.5311 (ns)
T9	132	0,307	0,1266	0,01202			
POS	66	0,3393	0,02079	0,02079			
FigSup3b							
	n	mean	SD	SEM			
U87 UBF-K	108	0,4351	0,1775	0,01708			
U251 Fib-K	86	0,369	0,3013	0,03249			

Therefore, we disagree, as our data conclusively that KANSL2 localizes in the nucleolus. The proposed alternative experiments (to demonstrate that KANSL2 localizes at the nucleolus) using RNA-FISH for 45SrRNA, besides being technically challenging, would not provide additional data than what we have already demonstrated with our ACTD assay and IF.

5. Line#331: Fig.3: Again KANSL2 staining from the representative images shows that this is highly diffuse. Furthermore, the extent of colocalization is also noisy and variable? for instance are the green (UBF) and red (KANSL2) compared across each of the sections across the confocal stacks? the images show that UBF and KANSL2 staining is rather different and is altered through cell cycle. Fig#3C: It is hard and extremely difficult to conclude from a single WB image, what are the levels of KANSL2 across independent biological replicates?

Reply: Thank you for your comment. KANSL2 signals are expected at both the nucleolus and nucleoplasm, as shown in Figures 2 and 3. We agree that the images from the original Figure 3 show that UBF and KANSL2 staining intensities were rather variable through the cell cycle phases. This is because the IF can produce differences in intensities, which is why we performed cell staining at the same time after harvesting them, to minimize this variability. We looked for more representative images and replaced them with a new Figure 3. Taking images of Z sections would have created an excessively large data volume for the synchronized cells IF. This would have made analysis difficult, so we did not pursue this option. The synchronized cells were imaged in a single focal plane, where clearly showing the nucleoli as black holes in DAPI. We evaluated red and green channel signals, calculating pixel intensities for those nucleolus regions. Previously called the 'colocalization value,' this term may have caused confusion. To clarify, it refers to the overlap between red (KANSL2) and green (UBF) signals in nucleoli, measured as correlated pixel intensities. We now refer to it as 'Correlation coefficient' (in nucleolus localized signals between red and green channels) in all quantification measures.

We agree that the UBF signals varied slightly through different phases of the cell cycle, although we did not find any evidence about this in the publications. However, in different experiments, we observed that KANSL2 signals in T3 and T6 were reduced compared to T0 and T9, suggesting the KANSL2 nucleolar localization is dynamic, as we described.

It is important to mention that although there are no differences in the correlation coefficient between T3 and T6, we observed that KANSL2 labeling in some cells at T6 is observed perinucleolar. This KANSL2 labeling could indicate the transition to the nucleolus in these cells to access T9. Future experiments will be necessary to confirm these observations.

Figure 3. Nucleolar localization of KANSL2 in synchronized cells. a) Immunofluorescence detection of endogenous KANSL2 and nucleolar markers UBF in HeLa cells. KANSL2 is located in different cellular compartments including the nucleolus. DAPI stained the nuclei. Adjustments to individual channels were performed to better visualize the merged images. Scale bar, 50 μ m. Cell cycle determined by analysis of synchronized cells stained with propidium iodide with flow cytometry. b) Correlation coefficient (i.e Pearson's correlation coefficient of pixels intensity) (N=2; total number of cells for T0=57, T3=102, T6=111, T9= 132, Negative control (NP) =132 and Positive Control=66 cells, respectively). For analysis, the raw images were used. ANOVA followed by Dunnett's test **** p <0.0001. c) Western blot analysis revealing KANSL2 protein levels in cell cycle stages. ANOVA followed by Turkey's test (n = 3).

More importantly, now we included in the Rebuttal (see below) and in the Figure 3c the semiquantification showing there is no changes in KANSL2 levels along the cell cycle.

Western blot analysis revealing KANSL2 protein levels in cell cycle stages.

6. Figure#4: From the image provided, it is rather hard to discern the localization of KANSL2 with the nucleolus. KANSL2 staining needs to be accompanied by a *bona fide* nucleolar marker/staining. Additional assays of collecting cells at each phase of the cell cycle and performing careful IF along with quantifications and/or western blot analyses of the lysates would lend more credence to this data. What is the effect of FBL or UBF knockdown on the nucleolar localization of KANSL2 and or across cell cycle? I suggest testing the effect of Cyclin/CDK inhibitors that perturb cell cycle and then assessing the staining/levels of KANSL2, FBL, UBF to more firmly establish the association of KANSL2 with cell cycle if any.

Reply: We respectfully believe that the reviewer's concern may stem from a misunderstanding of the images presented. In Figure 4, KANSL2-RFP clearly localizes within the nucleolar regions, identifiable as the DAPI-negative "black holes." To confirm this localization, we co-expressed GFP-p14ARF, a well-established nucleolar marker that labels cells with intact nucleolar activity (Wang et al., *Mod Pathol.* 2005, Lindström et al., *Exp Cell Res.* 2000). The two signals show almost complete overlap, confirming the nucleolar localization of KANSL2-RFP.

We appreciate the reviewer's suggestions regarding additional assays. However, we believe that the proposed experiments (FBL or UBF knockdown, or treatments with Cyclin/CDK inhibitors) would not substantially extend the current findings on nucleolar localization, as KANSL2-RFP is expressed under a constitutive promoter and remains stable throughout the cell cycle. Instead, we addressed the cell-cycle dependence of KANSL2 localization, as shown in the original Fig. 4, by co-expressing GFP-hPCNA, which displays characteristic distribution patterns across the different phases of the cell cycle (Essers et al., *Mol Cell Biol* 2005; Somanathan et al., *J Cell Biochem* 2001; Leonhardt et al., *J Cell Biol* 2000, Schönenberger, F et al 2001, Barr et al 2016, Zerjatke, T et al 2017). These results clearly demonstrate that KANSL2-RFP localization varies dynamically during the cell cycle.

- Wang JL, Zheng BY, Li XD, Nokelainen K, Angström T, Lindström MS, Wallin KL. p16INK4A and p14ARF expression pattern by immunohistochemistry in human papillomavirus-related cervical neoplasia. *Mod Pathol.* 2005 May;18(5):629-37 doi: 10.1038/modpathol.3800308. PMID: 15502810.

- Lindström MS, Klangby U, Inoue R, Pisa P, Wiman KG, Asker CE. Immunolocalization of human p14(ARF) to the granular component of the interphase nucleolus. *Exp Cell Res*. 2000 May 1;256(2):400-10. doi: 10.1006/excr.2000.4854. PMID: 10772813.

- Essers J, Theil AF, Baldeyron C, van Cappellen WA, Houtsmuller AB, Kanaar R, Vermeulen W. Nuclear dynamics of PCNA in DNA replication and repair. *Mol Cell Biol*. 2005 Nov;25(21):9350-9. doi: 10.1128/MCB.25.21.9350-9359.2005. PMID: 16227586; PMCID: PMC1265825.

- Somanathan S, Suchyna TM, Siegel AJ, Berezney R. Targeting of PCNA to sites of DNA replication in the mammalian cell nucleus. *J Cell Biochem*. 2001;81(1):56-67. doi: 10.1002/1097-4644(20010401)81:1<56::aid-jcb1023>3.0.co;2-#. PMID: 11180397.

- Leonhardt H, Rahn HP, Weinzierl P, Sporbert A, Cremer T, Zink D, Cardoso MC. Dynamics of DNA replication factories in living cells. *J Cell Biol*. 2000 Apr 17;149(2):271-80. doi: 10.1083/jcb.149.2.271. PMID: 10769021; PMCID: PMC2175147.

- Schönenberger, F., Deutzmann, A., Ferrando-May, E., & Merhof, D. Discrimination of cell cycle phases in PCNA-immunolabeled cells. *BMC Bioinformatics* 16:180 (2015). <https://doi.org/10.1186/s12859-015-0618-9>

- Barr, A. R., et al. A Dynamical Framework for the All-or-None G1/S Transition. *Cell Syst* 2:27–37 (2016). <https://doi.org/10.1016/j.cels.2016.01.001>.

- Zerjatke, T., et al. Quantitative Cell Cycle Analysis Based on an Endogenous All-in-One Reporter for Cell Tracking and Classification. *Cell Rep* 19:1953–1966 (2017). <https://doi.org/10.1016/j.celrep.2017.05.022>

7. Fig#5 and 6: While these assays are interesting and potentially suggest the involvement of KANSL2 in the regulation of RNA biogenesis and cell proliferation, would be useful to attempt rescue assays using full-length constructs of KANSL2 along with specific deletion or point mutations as suggested above to address the specificity of the role of KANSL2 in RNA biogenesis, cell proliferation. These assays show strong correlation but are unfortunately not conclusive enough to arrive at their purported roles in RNA biogenesis/function. It is important to note that even minor perturbations of a multitude of nuclear or cellular proteins can impact nucleolar transcripts, since the nucleolus is a major stress sensing sub-organelle.

Reply. Considering both effects, KD and Overexpression, we can conclude KANSL2 has role in rRNA transcription. In fact, the DNA chip and puromycin results support this conclusion. However, to address the specificity of the role of KANSL2 in rRNA biogenesis and to further support our conclusion, we have performed the rescue assay as suggested by this reviewer, confirming the role of KANSL2 in ribogenesis and proliferation (please see fig at the beginning of this rebuttal, Suppl Fig S7 a and b).

8. Fig. 7: While it is interesting to note that there is an effect of KANSL2 KD on genes regulating ribosome biogenesis, rescue assays would be useful to determine if overexpression of KANSL2 in an KANSL2 depleted background would rescue this effect? Is the expression dysregulation similar in cells cultured as a monolayer as opposed to a spheroid?

Reply. We agree with the reviewer's comment and have now included the requested rescue assays (see response to comment 7). Re-expression of full-length KANSL2-RFP in KANSL2-depleted cells successfully rescued the knockdown phenotype, restoring both proliferation and rRNA synthesis (please see above).

Most of our experiments were conducted in monolayer cultures, whereas the RNA-seq analysis was performed in 3D spheroid conditions. Although we did not specifically compare the effects of KANSL2 under different culture conditions, our transcriptomic data indicate that KANSL2 is required for the proper expression of several ribosomal proteins and factors involved in ribosome biogenesis in spheroid-grown cells.

9. While I understand that this work focuses on GBM cells and GBM as a system to establish the role of KANSL2, the role of KANSL2 could be tested on a few other cell types to address if the effects of KANSL2 are generalizable or cell-type/tissue type specific.

Reply: We speculate that the effect of KANSL2 on ribogenesis is more generalized than in GBM, as we observed KANSL2 nucleolar localization in other cell types (HeLa, HEK, and Neural progenitor cells, Fig. 3 and Supp Fig. S2) in addition to commercially available and established patient-derived GBM stem cell lines. Moreover, KANSL2-KD HeLa cells also decreased protein synthesis (Supplemental Fig. S6), indicating a direct effect on rRNA and ribogenesis..

Supplemental Figure S6. a) Western blotting analysis of KANSL2, KANSL1, KANSL3, MOF, RPS3, RPS18 and GAPDH levels after KANSL2 KDs in U87 cells. b) Representative confocal images and quantification of puromycin signals in U87 (number of cells = 43 nt y 70 siRNA KANSL2) and HeLa (number of cells = 54 nt y

83 siRNA) cells KANSL2-KD cells (siRNA for 48 h) treated as indicated and incubated with puromycin antibody. t-test Scale bar, 20 μm .

Minor comments and edits:

1. The introduction can be written much better. There is a frequent transition between concepts. It would be more engaging to read one concept and then link this to the next concept, so that the reading is seamless and flows well.

Reply: We thank the reviewer for this constructive comment. We have carefully revised the Introduction to improve clarity and readability by refining the transitions between concepts and ensuring a more logical flow of ideas. While certain topics need to be presented in a specific order to provide the necessary background, we have made targeted modifications to strengthen the narrative and create smoother connections between sections. For example, we moved the paragraph describing KANSL2 to the second paragraph (Page 3, line 74).

2. Fig#2: Conventionally, DAPI channel is arranged to the extreme left followed by red, or green and merged.

Reply: We thank the reviewer for this constructive comment. We now reorganized the DAPI panel to the left of each figure.

3. Materials and methods; Line#133; The bioinformatics assay is rather unclear, also, would be useful to mention the statistical analyses performed?

Reply: Thank you. We have mention statistical analysis details as you requested in the corresponding section (Page 5, line 137).

4. Immunofluorescence assays: how many cells were grown and for how long? What was the pH of each of the solutions used? Image acquisition parameters: what was the X-Y frame size? voxel size?

Reply: Thank you for the feedback. We have included the missing data for the quantity of cells and pH of the used solutions in the corresponding section.

Image acquisition parameters were the following:

U87: X-Y frame size (image size) : 212.47 μm x 212.47 μm . x-y: 0.079 μm /z: 1.1 μm . voxel size: 0.079 μm x 0.079 μm x 1.1 μm .

U251: X-Y frame size (image size): 212.47 μm x 212.47 μm . x-y: 0.079 μm /z: 1.1 μm . voxel size: 0.079 μm x 0.079 μm x 1.1 μm .

Hela: X-Y frame size (image size) : 212.47 μm x 212.47 μm . x-y: 0.079 μm /z: 1.1 μm . voxel size: 0.079 μm x 0.079 μm x 1.

We are submitting the original images obtained from Zeiss microscopy converted it into a high quality tiff file to upload to the system following the editor suggestion so the reviewer is able to revise the frame size and other parameters.

5. Transfections: What was the volume of Lipofectamine 3000 used? and for how many cells?

Reply: We thank the reviewer for this observation, which helped us clarify the conditions of our transfection experiments. We incorporated the missing info in the text “For each well, 3.75 μ l of Lipofectamine 3000 reagent was mixed with 2 μ g of plasmid DNA and 4 μ l of P3000 reagent in Opti-MEM reduced-serum medium (Thermo Fisher Scientific)” (Page 7, line 199).

6. shRNA and siRNA conditions are incomplete, what was the concentration of the shRNA plasmid, siRNA for transfections?

Reply: We thank the reviewer for this observation, which helped us to clarify our knockdown protocols. We have now specified the conditions used, including the amount of shRNA plasmid DNA, the siRNA concentrations, and the volume of Lipofectamine RNAiMAX employed. We included the details in the text “Cells were co-transfected with 1 μ g of specific shRNA plasmid, 2 μ g of pMD2.G (VSVG envelope plasmid), and 2 μ g of psPAX2 (packaging plasmid, D8.9) using polyethyleneimine (PEI) at a DNA:PEI ratio of 1:6.” (Pages 7 and 8, lines 196-233).

7. pH of the buffers must be mentioned as frequently as possible for each of the methods. (Nico)

Reply: We thank the reviewer for this helpful suggestion. We have revised the Materials and Methods to include the pH values of all buffers used across our experimental procedures (cell culture, immunofluorescence, transfections, shRNA/siRNA knockdown, RNA extraction/qPCR, ChIP-qPCR, western blotting, luciferase assays, and proliferation assays). This additional information ensures clarity and reproducibility of the described methods. You will find them in the corresponding section (pages 4 -11).

8. Molecular weight markers are missing for most of the WB in the supplementary figures.

Reply: we are submitting as supplemental information for the blots figures those original membranes with molecular weight markers indicated.

These are some examples.

Reviewer #2 (Remarks to the Author):

The authors have tried to investigate the role of KANSL2, a subunit of the non-specific lethal (NSL) chromatin-modifying complex, in glioblastoma (GBM). The authors demonstrate that KANSL2 is overexpressed in GBM (compared to normal brain tissue) and is positively correlated with stemness and ribosomal protein gene expression. The protein localizes dynamically to the nucleolus, and its overexpression enhances, while its knockdown suppresses, rRNA transcription and cell proliferation. However, a major message from this paper regarding the regulation of KANSL2 and its role in stemness has been published by the same group previously, which they have acknowledged generously as ref. 19 in the manuscript. Apart from this, the only aspect of novelty is the “ribosomal” involvement where again it is not clear, whether the focus of authors is “ribosomal RNA (rRNA) or ribosomal proteins (ribosomal protein mRNA)” and this needs further experimentation to sound credible.

While the manuscript provides intriguing observations, several major and minor issues require clarification or additional experimentation to support the conclusions.

Reply: The main focus of this study is not the regulation of stemness by KANSL2, which we previously reported (Ref. 19, now reference 12), but rather its novel role in ribosome biogenesis. We have now revised several sections of the manuscript to clarify this point and to strengthen the description of our new findings.

Specifically, we provide evidence that KANSL2 regulates both rRNA synthesis and the expression of ribosomal protein genes, indicating a broader role for this factor in coordinating ribosome biogenesis and cell proliferation in GBM cells. We believe these results expand the current understanding of KANSL2 function and represent a distinct contribution from our earlier work.

Major Comments

1. Redundancy of Initial Observations:

a) The findings in Figure 1a and 1b replicate previously published data by the same group (ref. 19), particularly the association of KANSL2 with stemness. Presenting this as a main figure seems unjustified unless new insights are added.

Reply: Thank you for the comment. Regarding the possible redundancy of our first result, as we explained, we have increased the size of the samples analyzed, which come from a public and independent database, different from the original ref 19. Thus, we confirmed undoubtedly that KANSL2 is increased in GBM samples and correlates with stemness by analyzing the whole transcriptome of GBM samples. I think these new analyses are worth including in the article. However, we agree to move these figures to the supplemental data.

Supplemental Figure S1: **a)** *KANSL2* expression of GTEX brain normal tissue samples vs TCGA-GBM tumor samples. t-test. **** $p < 0.001$. **b)** Spearman's correlation and linear regression analysis of *KANSL2* expression versus stemness index score ($n = 166$; $p < 0.0001$). **c)** Transcriptomic analysis of NSL complex members (*KANSL1*, *KANSL3*, *WDR5*, *MCRS1* and *KAT8*) in TCGA-GBM cancer. A column scatter plot showing the normalized ($\text{log}_2(\text{normcount} + 1)$) mRNA expression of NSL complex members in GTEX brain normal tissue ($n = 1136$) versus TCGA-GBM primary tumor ($n = 166$). t-test, **** $p \leq 0.0001$.

b) Moreover, the method of patient stratification based on *KANSL2* expression for differential gene expression (DGE), GSEA, and GO analyses lacks clarity. The thresholding approach for "high" vs. "low" *KANSL2* expressers must be described in detail. Was a statistical method (e.g., median split, tertiles, or clustering) employed, and is it biologically meaningful or relevant? Also, Fig. 1a does not indicate a bimodal expression distribution, further complicating justification for this classification.

Reply: Thank you for the comment. We have now added more details to the bioinformatics analysis we performed. We have specified the statistical method employed in the text (Page 5).

For the stratification of patients into "high" and "low" *KANSL2* expression groups, we applied a mean expression split. While *KANSL2* expression in TCGA-GBM does not display a clear bimodal distribution, dichotomization by mean expression is a widely used approach in transcriptomic and survival analyses when continuous expression values need to be converted into groups for DGE, GSEA, or GO analysis. This method provides balanced group sizes, increases statistical power, and avoids bias toward extreme subgroups that may occur with tertile or quartile splits, particularly in datasets with moderate sample sizes.

Importantly, our choice is biologically meaningful: using the mean threshold allowed us to capture genes and pathways differentially regulated across the continuum of KANSL2 expression, rather than restricting the analysis to outlier patients. This approach has been applied in several TCGA-based studies where candidate gene expression lacks bimodal distribution but is still functionally relevant.

2. Ambiguity in Localization Claims:

a) The conclusion that actinomycin D (ACTD) treatment leads to a loss of nucleolar localization is premature unless total KANSL2 expression (both RNA and protein) is shown to remain unaffected by treatment. A decrease in signal could simply reflect downregulation, not redistribution.

Reply: We thank the reviewer for this comment. To address this concern, we analyzed KANSL2 protein levels after ACTD treatment by western blot, followed by densitometric quantification of band intensity (Fig. 2d and see below). The results show no significant change in KANSL2 expression upon treatment. Therefore, the observed reduction in nucleolar signal reflects a relocalization of KANSL2 rather than a downregulation of its expression.

Western blot and quantification of KANSL2 by densitometry after ACTD treatment.

b) Similarly, the interpretation of shRNA-mediated knockdown as a loss of nucleolar localization is flawed. A global decrease in KANSL2 across all compartments is expected with knockdown, and cannot be interpreted as redistribution. Figure S3b clearly suggests a uniform reduction rather than specific relocalization.

Reply: We thank the reviewer for this clarification. Indeed, shRNA-mediated knockdown of KANSL2 resulted in a global reduction of KANSL2 signal across all cellular compartments, including the nucleolus, as expected for an effective depletion (now Suppl Fig. S4). This confirms that the nucleolar signal observed under control conditions corresponds to endogenous KANSL2 and is not an artifact of immunofluorescence. We did not intend to interpret these results as evidence of relocalization.

Supplemental Figure S4. a) Representative image of western blot analysis showing KANSL2 protein levels in U87 stably infected with two different KANSL2 shRNAs (I and II). b) Immunofluorescence detection of endogenous KANSL2 and nucleolar UBF in U87 cells KANSL2-depleted cells indicating a general reduction of KANSL2 expression signal.

3. Cell Cycle–Dependent Localization:

The proposed relationship between KANSL2 nucleolar localization and the cell cycle remains vague. The authors should:

a. Quantify KANSL2 levels in nuclear vs. cytoplasmic/nucleolar fractions across different cell cycle phases.

Reply: Our results already provide consistent evidence of a cell cycle–dependent nucleolar localization of KANSL2. A quantitative comparison of KANSL2 levels in cytoplasmic, nuclear, and nucleolar fractions could certainly add complementary information; however, such experiments are technically difficult to perform with precision. The nucleolus is a membrane-less and highly dynamic structure whose isolation by standard sonication and sucrose-gradient methods often disrupts its phase-separated organization, leading to loss of integrity and cross-contamination with other nuclear components. For these reasons, we did not attempt biochemical nucleolar fractionation in this study. Instead, we relied on imaging-based analyses (Fig. 3 and 4), which reproducibly show that KANSL2 localization within the nucleolus varies dynamically throughout the cell cycle.

b. Revisit the assertion in line 358 linking cell cycle phase (particularly S-phase) with rRNA biogenesis via KANSL2.

Reply: In line 358, it can read: "Some nucleolar proteins involved in ribosome biogenesis, such as Nucleolin, RPS6, and c-Myc shuttle between nucleus and cytoplasm during the cell cycle[51–53], which might suggest that KANSL2 behaves in a similar way during rRNA biogenesis." We

noticed we have mistyped and this sentence should be read “between nucleolus, nucleus and cytoplasm” (Page13, Line 416).

In this sentence, we wanted to emphasize that KANSL2 can go in and out of the nucleolus dynamically, as was described for other proteins involved in ribogenesis. We also replaced "in a similar way" with "similarly".

c. Clarify whether KANSL2 localization changes precede or follow changes in rRNA expression.

Reply: This is an interesting question, although it is technically very challenging to answer. Nevertheless, our findings suggest that KANSL2 may precede the initiation of rRNA transcription. Specifically, KANSL2 acts through epigenetic modification, as clarified by a DNA chip assay, and nucleolar localization found in late G1 (or G1/S) and G2, a cell phase during which the literature has detected acetylation of H4 in rDNA and nascent rRNA after mitosis (Brown SE et al 2008, Klein J et al 1999).

- Brown SE, Szyf M. Dynamic epigenetic states of ribosomal RNA promoters during the cell cycle. *Cell Cycle*. 2008 Feb 1;7(3):382-90. doi: 10.4161/cc.7.3.5283. Epub 2007 Nov 5. PMID: 18235221.

- Klein J, Grummt I. Cell cycle-dependent regulation of RNA polymerase I transcription: the nucleolar transcription factor UBF is inactive in mitosis and early G1. *Proc Natl Acad Sci U S A*. 1999 May 25;96(11):6096-101. doi: 10.1073/pnas.96.11.6096. PMID: 10339547; PMCID: PMC26841.

We modified the discussion to clarify this point (Page 16 Line 512-521).

“Remarkably, overexpressing KANSL2-RFP resulted in partial localization in the nucleolus. This also increased steady-state rRNA biosynthesis. In contrast, silencing KANSL2 decreased rRNA biosynthesis. A luciferase activity reporter assay (using ribosomal rDNA-Luc promoter) confirmed that KANSL2 affects transcription initiation. Moreover, reduced acetylated levels of H4 at lysine 5 and 8 (H4K5ac and H4K8ac) were observed in the rDNA promoter region in KANSL2-silenced cells. This suggests the NSL complex regulates rRNA expression epigenetically. This finding aligns with the known epigenetic status of rRNA genes during the cell cycle in synchronized HeLa cells. Acetylation of rDNA promoter-associated histones and transcription of rRNA primarily occur in late G1 and G2⁷¹. These are the time frames in which we observed KANSL2 localization in the nucleolus. This suggests KANSL2 is required for, and may precede, the initiation of rRNA transcription. In summary, these studies suggest that KANSL2 mediates ribogenesis in GBM.”

d. Notably, the lack of change in total KANSL2 levels during the cell cycle makes this hypothesis speculative without subcellular fractionation data.

Reply: We attempted nucleolar fractionation to complement our imaging data, but the results were inconsistent due to the intrinsic limitations of current protocols. Standard purification methods involving sonication and sucrose-gradient centrifugation are not optimal for preserving nucleolar integrity. The nucleolus is a membrane-less organelle composed of coexisting liquid phases (DFC and GC) maintained by weak molecular interactions and extremely low surface tension ($\approx 10^{-6}$ – 10^{-7} N/m; *Feric et al., Cell* 2016; *Lafontaine et al., Nat Rev Mol Cell Biol* 2021). The mechanical stress applied during sonication exceeds these weak cohesive forces, leading to disruption of the condensate and recovery of unstructured molecular components rather than intact nucleoli.

For these reasons, we focused on immunofluorescence analyses, which clearly show KANSL2 signal within nucleoli under native conditions. These imaging-based experiments consistently demonstrate that KANSL2 nucleolar localization is dynamic throughout the cell cycle phase. In addition, total KANSL2 protein levels were quantified by densitometric analysis and showed no significant differences between the evaluated time points ($n = 3$ independent experiments; Fig 3c).

-Lafontaine, D.L.J., Riback, J.A., Bascetin, R. *et al.* The nucleolus as a multiphase liquid condensate. *Nat Rev Mol Cell Biol* **22**, 165–182 (2021). <https://doi.org/10.1038/s41580-020-0272-6>

-Feric M, Vaidya N, Harmon TS, Mitrea DM, Zhu L, Richardson TM, Kriwacki RW, Pappu RV, Brangwynne CP. Coexisting Liquid Phases Underlie Nucleolar Subcompartments. *Cell*. 2016 Jun 16;165(7):1686-1697. doi: 10.1016/j.cell.2016.04.047. Epub 2016 May 19. PMID: 27212236; PMCID: PMC5127388.

Western blot analysis revealing KANSL2 protein levels in cell cycle stages.

4. Selective rRNA/Ribosomal Upregulation:

The upregulation of 45S pre-rRNA upon KANSL2 overexpression without a corresponding increase in mature rRNA species (e.g., 18S, 5.8S) is puzzling. The authors should investigate and explain:

a. Why only a subset of rRNA or ribosomal proteins (e.g., only RPS18 in Fig. S5) are affected.

Reply: We thank the reviewer for the comment. This remains unclear and will require further study. Our results indicate that KANSL2 increases rRNA transcription. However, the evidence also indicates that KANSL2 levels affect the expression of r-proteins and factors involved in the processing and assembly of the 60S and 40S. We propose that KANSL2 could impact the amount of pre-rRNA, and as the reviewer suggested, the differences in the rRNA amount could be the result of incomplete processing or post-transcriptional modifications. Although finding the mechanism is out of the scope of this manuscript, we now add a paragraph discussing about this.

We observed that KANSL2 depletion decreases 45S and 28S rRNA levels, whereas 18S rRNA remains unchanged. This suggests that KANSL2 does not globally regulate rRNA synthesis, but rather affects specific steps of ribosome biogenesis (now discussed in Page 16, line 519).

“Ribosomal RNA (rRNA) processing is a vital cellular process and recent studies have shown that is spatially segregated within the nucleolus: early processing events leading to 18S (SSU) formation occur in the dense fibrillar component (DFC), while later stages generating 28S/5.8S (LSU) take place in the granular component (GC)⁷². Also, pre-rRNA processing and maturation regulate the shape and phases of compartmentalization observed in the nucleolus⁷². Our results indicate that KANSL2 levels also regulate the expression of enzymes and factors (mainly r proteins) required for proper ribosome processing and assembly. Particularly, KANSL2 KD caused a differential accumulation of 45S and 28S rRNA, accompanied by unregulated rRNA protein expression. Potentially, KANSL2 could specifically influence LSU-related processing or assembly factors located in the GC, which could explain why KANSL2 affected those rRNAs. Further experiments will be required to explore this.”

Recent work has shown that rRNA processing is spatially segregated within the nucleolus: early processing events leading to 18S (SSU) formation occur in the dense fibrillar component (DFC), while later stages generating 28S/5.8S (LSU) take place in the granular component (GC) (Quinodoz et al., Nature 2025). Therefore, KANSL2 may specifically influence LSU-related processing or assembly factors localized in the GC as we suggested above.

Quinodoz, S.A., Jiang, L., Abu-Alfa, A.A. et al. Mapping and engineering RNA-driven architecture of the multiphase nucleolus. *Nature* 644, 557–566 (2025). <https://doi.org/10.1038/s41586-025-09207-4>.

b. Whether this implies incomplete processing of rRNA or post-transcriptional bottlenecks

Reply. This comment is relevant, and we appreciate it. Likewise, we consider it could be a processing and/or post-transcriptional bottleneck, but this question is beyond our current scope.

However, as mentioned above, KANSL2 levels may differentially affect the biogenesis of the 18S/40S; it would be relevant to investigate in the future. We modified the Fig 7 adding the name of the ribosomal genes in the volcano plot.

Figure 7c) Volcano plot of differentially expressed genes in the KANSL2 KD GBM derived samples (LFC = log₂ fold change).

We have also extended the discussion about the potential mechanism of these processing defects mediated by KANSL2 KD as described above.

The synthesis of ribosomal RNA (rRNA) is accompanied by many intermediates that are difficult to isolate and characterize. There is a surveillance mechanism ensuring stoichiometry of ribosome proteins (Robledo et al 2018), and probably also, there would be a similar mechanism to ensure appropriate levels of rRNAs and pre-rRNA. Our results indicate that KANSL2 levels regulate the expression of enzymes and factors (mostly r-proteins) required for the correct processing and assembly of 18S/40S. KANSL2 knockdown may cause differential accumulation of rRNAs since unregulated expression of r-proteins unbalances the factors for proper ribosome assembly.

-Robledo S, Idol RA, Crimmins DL, Ladenson JH, Mason PJ, Bessler M. The role of human ribosomal proteins in the maturation of rRNA and ribosome production. *RNA*. 2008 Sep;14(9):1918-29. doi: 10.1261/rna.1132008. Epub 2008 Aug 12. PMID: 18697920; PMCID: PMC2525958.

c. Additional evidence (e.g., northern blot or qPCR of intermediate species) could help resolve this.

Reply. We do not attempt to determine the mechanism that would explain the selective rRNA/Ribosomal upregulation/downregulation caused by KANSL2 in this work. This would be beyond our scope. Northern blot or qPCR, we will only reveal what intermediate species are (or not), enforcing our findings already described in the results.

5. Potential Contradiction in Cell Cycle Data:

The data presented in Figures 3, 4c, and 6c raise interpretational challenges:

a. KANSL2 is reported to be low in the S-phase nucleolus.

Reply: Yes, we describe a lower KANSL2 accumulation in the nucleolus mostly in mild and late S-phase in synchronized cells. Since KANSL2 expression levels do not appear to vary throughout interphase, one might speculate that the remaining protein is directed to other organelles, such as mitochondria, to perform other functions also related to the S phase and cell cycle progression in addition to the nuclear function.

It is important to mention that although there are no differences in the correlation coefficient between T3 and T6, we observed that KANSL2 labeling in some cells at T6 is observed perinucleolar. This KANSL2 labeling could indicate the transition to the nucleolus in these cells to access T9. Future experiments will be necessary to confirm these observations.

Figure 3. Nucleolar localization of KANSL2 in synchronized cells. a) Immunofluorescence detection of endogenous KANSL2 and nucleolar markers UBF in HeLa cells. KANSL2 is located in different cellular compartments including the nucleolus. DAPI stained the nuclei. Adjustments to individual channels were performed to better visualize the merged images. Scale bar, 50 μ m. Cell cycle determined by analysis of synchronized cells stained with propidium iodide with flow cytometry. b) Correlation coefficient (i.e

Pearson's correlation coefficient of pixels intensity) (N=2; total number of cells for T0=57, T3=102, T6=111, T9= 132, Negative control (NP) =132 and Positive Control=66 cells, respectively). For analysis, the raw images were used. ANOVA followed by Dunnett's test ****p≤0.0001. c) Western blot analysis revealing KANSL2 protein levels in cell cycle stages. ANOVA followed by Turkey's test (n = 3).

b. Yet, KANSL2 knockdown reduces the S-phase population.

Reply: Yes, KD reduces all KANSL2 present in the cell, suggesting KANSL2 in G1 facilitates to enter into S-phase. These two observations are not contradictory. The lower nucleolar accumulation of KANSL2 during mid- and Late-S phase likely reflects a transient redistribution of the protein rather than a decrease in its total cellular levels, consistent with the dynamic organization of the nucleolus during DNA replication. In line with this, KANSL2 knockdown reduces the S-phase population, suggesting that KANSL2 activity in G1 is required for proper entry into S phase. Therefore, the reduced nucleolar signal observed during S phase and the decrease in S-phase cells upon knockdown are fully compatible and likely reflect distinct aspects of KANSL2's cell cycle-dependent functions.

c. This appears constitutive—does KANSL2 promote or suppress S-phase entry?

Reply: Most of the KD cells were in the non-S phase. Therefore, KD of KANSL2 may suppress S-phase entry.

Supplemental Figure S7 d) Cell-cycle analysis of U87 cells after KANSL2 depletion, determined by propidium iodide staining and flow cytometry. The percentage of cells in G0/G1, S, and G2/M phases is shown. Data are shown as mean ± SD (n = 3). One-way ANOVA followed by Dunnett's test; ****p ≤ 0.0001.

d. The authors should clarify this relationship, potentially exploring whether nucleolar KANSL2 levels act as a checkpoint or modulator for S-phase progression.

Reply: this is an interesting question, but we don't know the answer. This question about KANSL2 (check point or modulator) is beyond the current scope of our article.

Minor Comments

1. Western Blot Quantification: All western blot results (particularly Fig. 5d) must be quantitatively analyzed with densitometry, normalized to loading controls (e.g., GAPDH). As it stands, the observed changes are marginal and may not be statistically significant.

Reply. We have done now.

Figure 5. **d)** Western blot analysis and quantification of puromylylated peptides under KANSL2 siRNA knockdown.

2. Redundant Literature Findings: Results presented in Figures 6a and 6b have been reported in previous literature, as acknowledged by the authors. These should be moved to the supplementary section to preserve the novelty and focus of the main text.

Reply: we moved them to supplemental.

Supplemental Figure S7 **b**) Crystal violet proliferation assay in U87 cells with stable KANSL2 knockdown or overexpression. Data are shown as mean \pm SD (n = 3). One-way ANOVA followed by Dunnett's test; ***p \leq 0.001

3. Inconsistency in Knockdown Effects: The phenotypic outcomes of KANSL2 knockdown using shRNA I and II (Fig. 6c) show considerable variability. The discrepancy is large enough that statistical comparison between the two might reveal significant differences. The authors should address:

Reply: The apparent variability between shRNA I and II reflects differences in silencing efficiency rather than inconsistency in the knockdown effects. Such differences are expected, as shRNA potency can vary depending on sequence and experimental context. In our experiments, both shRNAs consistently reduced KANSL2 expression and produced comparable phenotypic outcomes, with the magnitude of the effects correlating with the relative strength of each construct. We have also updated the corresponding graph to better represent the data and clarify the scale of variation between shRNA I and II. Suppl Fig 7d (please see above).

a. The knockdown efficiency of both constructs.

Reply: you can find it in the Fig 5b.

b

b. Whether off-target effects or differential knockdown kinetics could explain this variation.

Reply: We have employed non-targeting (NT) vector as control as well as different shRNA sequences indicating the off-target chance is very low (please see the figure in response to 2b above).

c. Replicating the experiments with rescue constructs could help validate specificity.

Reply. We agree with this comment. We performed this experiment. We observed that KANSL2 forced-expression in KANSL2-KD cells restored the proliferation as well as the synthesis of 45S and 28S without generating changes in 18S.

These results, as suggested by the reviewer, indicate that the specific effects are mediated by KANSL2 expression.

Supplemental Figure S7

Supplemental Figure S7 a) RT-qPCR analysis of 45S pre-rRNA, 28S, and 18S rRNA levels in U87 cells expressing the indicated constructs: non-targeting control (NT) + RFP, NT + KANSL2-RFP, shKANSL2-II + RFP, and shKANSL2-II + KANSL2-RFP. Re-expression of KANSL2-RFP restored rRNA synthesis to levels comparable to control cells in KANSL2 knockdown cells. Data are shown as mean \pm SD ($n = 3$). Two-way ANOVA followed by Fisher's least significant difference (LSD) post hoc test (uncorrected); * $p \leq 0.05$, *** $p \leq 0.001$. **b)** Crystal violet proliferation assay in U87 cells with stable KANSL2 knockdown or overexpression. Data are shown as mean \pm SD ($n = 3$). One-way ANOVA followed by Dunnett's test; *** $p \leq 0.001$. **c)** Crystal violet proliferation assay in U87 and U251 cells expressing the indicated constructs: NT + RFP, NT + KANSL2-RFP, shKANSL2-II + RFP, and shKANSL2-II + KANSL2-RFP. Re-expression of full-length KANSL2-RFP rescued the proliferation defect caused by KANSL2 depletion. Data are shown as mean \pm SD ($n = 3$). Two-way ANOVA followed by Fisher's LSD post hoc test (uncorrected); ** $p \leq 0.01$, *** $p \leq 0.001$, **** $p \leq 0.0001$. **d)** Cell-cycle analysis of U87 cells after KANSL2 depletion, determined by propidium iodide staining and flow cytometry. The percentage of cells in G0/G1, S, and G2/M phases is shown. Data are shown as mean \pm SD ($n = 3$). One-way ANOVA followed by Dunnett's test; **** $p \leq 0.0001$.

Reviewers' comments:

Reviewer #1 (Remarks to the Author):

While the manuscript has indeed improved after revision, my previous concerns regarding the clarity of the images of KANSL2 localization still remains as the localization is still rather heterogeneous and diffuse. In silico analyses and expression studies suggest the involvement of KANSL2 in regulating gene expression of ribosomal transcripts, the evidence is rather circumstantial in nature and does not quite address the specificity of KANSL2 localization in the nucleolus.

Reply:

We acknowledge the reviewer's concern regarding the apparent heterogeneity of KANSL2 nucleolar localization. While we agree that KANSL2 displays a variable subnuclear distribution, we are confident that the combined evidence presented (including bioinformatic, gene expression analysis, immunolocalization analysis, DNA chip analysis, and DNA reporter analysis) supports a specific and biologically meaningful association of KANSL2 with the nucleolus in the context of ribosome biogenesis.

We recognize that the nucleolar localization of KANSL2 appears heterogeneous across cells. As extensively reviewed by Iarovaia *et al.* (Iarovaia OV *et al.*, 2019), the nucleolus functions as a highly dynamic hub in which numerous factors shuttle in and out depending on cell-cycle stage, transcriptional demand, and cellular stress. In this context, the variable nucleolar enrichment of KANSL2 that we observe is consistent with the dynamic behavior described for other regulatory nucleolar proteins. Therefore, the heterogeneous pattern aligns with the current understanding of nucleolar biology.

Iarovaia OV, Minina EP, Sheval EV, Onichtchouk D, Dokudovskaya S, Razin SV, Vassetzky YS. Nucleolus: A Central Hub for Nuclear Functions. *Trends Cell Biol.* 2019 Aug;29(8):647-659. doi: 10.1016/j.tcb.2019.04.003. Epub 2019 Jun 5. PMID: 31176528.

Rev #1:1. Figure 2: Please provide zoomed in images of Fig. 2A or 2B, such that the localization of KANSL2 can be clearly discerned. Since the KANSL2 panel in Fig.2A is low magnification, the staining of KANSL2 (red) is unclear. I suspect this to be due to issues with permeabilization. Cells can be pre-permeabilized with CSK buffer to partially digest the cytoplasm, which would allow better permeability of both UBF and KANSL2 antibodies, as their access especially into the nucleus and nucleolus is particularly difficult.

Reply:

We appreciate the reviewer's concern regarding our immunolocalization protocol. As previously explained, our immunofluorescence protocol, staff expertise, and

established procedures have ensured minimal artifacts throughout the procedure. Furthermore, our standardized protocol incorporates optimal controls. We have considerable experience with this type of experiment, and issues related to permeabilization troubleshooting were addressed and dismissed. We would like to clarify that our immunofluorescence protocol includes well-established fixation and permeabilization steps that consistently preserve subnuclear structures. Importantly, the nucleolar markers used in these experiments (UBF and fibrillarin) display the expected strong and well-defined nucleolar signal across the cells. If the permeabilizations were suboptimal, these markers would also exhibit weak or inconsistent staining, which is not the case in our images. Therefore, we are confident that our images exhibit a lower risk of artifacts during acquisition and are of high quality and clarity.

We agree that higher-magnification views would facilitate visualization of KANSL2 subcellular distribution, and we have reorganized the panels in Figure 2 and added higher-magnification insets to help the reviewer and readers better appreciate KANSL2 nucleolar enrichment where present. These new insets allow clearer appreciation of the nucleolar enrichment observed in a subset of cells. Also, as previously, we are providing a detailed Z-sectioning of a single U251 cell (Supplemental Fig S2) in which KANSL2 nucleolar localization pattern can be seen in much more detail.

Figure 2.

Supplemental Fig S2b.

Rev #1: 2. Fig.2B shows KANSL2 localization potentially in the nucleolus of only a single cell, which I find particularly worrisome since if the staining was ubiquitous and consistent, this should have shown up across the nucleolus of all cells. I suspect that the relatively low correlation co-efficient of <0.2 as quantified in Fig;2c largely stems from the rather reduced localization of KANSL2 across cells in the nucleolus, possibly due to accessibility issues. Is this correct?

Reply:

We understand the reviewer's concern regarding the heterogeneity of KANSL2 nucleolar enrichment in Fig. 2B. As shown in later sections of the manuscript (Fig. 3), KANSL2 nucleolar localization is cell-cycle dependent, which naturally results in variable nucleolar signal intensity across the population. Thus, cells captured in different stages of the cycle display different degrees of nucleolar enrichment, accounting for the heterogeneity observed.

To avoid any potential misinterpretation, we have replaced the original panel with images that include a larger number of cells and now also provide zoomed-in insets, which allow clearer visualization of KANSL2 enrichment within nucleoli.

Regarding the correlation coefficient, a value in the ~ 0.2 range is fully compatible with the biology of a regulatory factor that only partially localizes to nucleoli and also resides in the nucleoplasm.

Rev #1: 3. Fig.2D: Furthermore, while the ActD treatment is appropriate, the already reduced/subdued localization of KANSL2, does not really leave much for the ActD to act upon? previous studies employing ActD treatment followed by nucleolin staining, shows a major disruption and dispersal of the nucleolus across the nucleoplasm.

Reply:

We thank you for your concern. In our hands, ACTD treatment robustly induced the canonical nucleolar stress response, indicating that the treatment was effective in

disrupting nucleolar organization. Specifically, UBF relocated to nucleolar caps, a well-established hallmark of Pol I inhibition and nucleolar segregation during transcriptional blockage (Shav-Tal et al., 2005). Importantly, we do not observe changes in total KANSL2 protein levels upon ACTD treatment (Fig. 2D), indicating that the reduction of nucleolar KANSL2 signal reflects **redistribution** rather than degradation or technical loss. Thus, the reduction of KANSL2 signals from the nucleolus under ACTD is consistent with nucleolar disassembly.

Shav-Tal Y, Blechman J, Darzacq X, Montagna C, Dye BT, Patton JG, Singer RH, Zipori D. Dynamic sorting of nuclear components into distinct nucleolar caps during transcriptional inhibition. *Mol Biol Cell*. 2005 May;16(5):2395-413. doi: 10.1091/mbc.e04-11-0992. Epub 2005 Mar 9. PMID: 15758027; PMCID: PMC1087244.

Rev #1: 4. Would be extremely useful to show zoomed inserts of the nucleus before and after KANSL2.

Reply: According to the editor, the reviewer kindly requests zoomed-in microscopy images of the nucleus (with staining for KANSL2) before and after KANSL2 knockdown. We have included new zoomed-in photos in the new Supplemental Figure S4.

Supplemental Fig. S4.

Rev #1: 5. Also, the manuscript would greatly benefit by some expert editing as I notice number of typographical and grammatical errors.

Reply: We appreciate the reviewer's comment. The manuscript has now been carefully revised for language, and typographical and grammatical errors have been corrected with the help of native English-speaking colleagues and co-authors, and are highlighted in the manuscript text file.